# Natural Products of Marine Macroalgae from South Eastern Australia, with Emphasis on the Port Phillip Bay and Heads Regions of Victoria

**DOI:** 10.3390/md18030142

**Published:** 2020-02-28

**Authors:** James Lever, Robert Brkljača, Gerald Kraft, Sylvia Urban

**Affiliations:** 1School of Science (Applied Chemistry and Environmental Science), RMIT University, GPO Box 2476V Melbourne, VIC 3001, Australia; james.lever@rmit.edu.au (J.L.); robert.brkljaca@monash.edu (R.B.); 2Monash Biomedical Imaging, Monash University, Clayton, VIC 3168, Australia; 3School of Biosciences, University of Melbourne, Parkville, Victoria 3010, Australia; gtk@unimelb.edu.au; 4Tasmanian Herbarium, College Road, Sandy Bay, Tasmania 7015, Australia

**Keywords:** marine macroalgae, bioactivity, secondary metabolites

## Abstract

Marine macroalgae occurring in the south eastern region of Victoria, Australia, consisting of Port Phillip Bay and the heads entering the bay, is the focus of this review. This area is home to approximately 200 different species of macroalgae, representing the three major phyla of the green algae (Chlorophyta), brown algae (Ochrophyta) and the red algae (Rhodophyta), respectively. Over almost 50 years, the species of macroalgae associated and occurring within this area have resulted in the identification of a number of different types of secondary metabolites including terpenoids, sterols/steroids, phenolic acids, phenols, lipids/polyenes, pheromones, xanthophylls and phloroglucinols. Many of these compounds have subsequently displayed a variety of bioactivities. A systematic description of the compound classes and their associated bioactivities from marine macroalgae found within this region is presented.

## 1. Introduction

The pharmaceutical industry has evolved as a result of research conducted in the areas of both synthetic organic chemistry and natural products extraction. During the period 1981–2014, approximately 42% of all U.S Food and Drug Administration (FDA) new drug approvals were based on either natural products or derivatives of a natural product pharmacophore [1]. Additionally, 49% of all anti-cancer drugs produced since the 1940s have been derived from a natural product source, or have been inspired by a natural product, and synthesized as a ‘natural product mimic’ [1].

Given the reduced effectiveness of traditional antibiotics to fight more resistant forms of bacterial infection in humans, together with the need for antibiotics in agriculture, there has been an increasing need to source new antibiotic drugs. Akin to this is the unabated need for bioactive compounds that show cytotoxic activity towards tumor cells for the effective treatment of cancers. This has provided much of the impetus for the research conducted within the field of natural product drug discovery, both from terrestrial and marine sources. While numerous drugs have been derived from terrestrial plants, there is still a huge untapped reserve of marine organisms that have been comparatively understudied. Terrestrial natural products (TNPs) have been exploited for their biological potency for many hundreds of years, whilst it is only recently, due to the increased use of SCUBA, that we have had access to the array of ocean-dwelling species, and this has led to an increase in the study of Marine Natural Products (MNPs). In fact, between 2014 and 2016, 203 new natural products were discovered from the study of macro algae (Green, Red and Brown) [2,3]. Many of these compounds displayed very promising biological activity, making them serious contenders as anti-cancer and anti-bacterial drugs or drug leads. A recent cheminformatics study has highlighted the potential of MNPs to produce drug-like chemical compounds [4]. Despite this, MNPs remain less studied than TNPs, mostly due to the relative ease with which TNP specimens can be obtained and cultivated. Currently, there are 10 FDA-approved marine-derived pharmaceutical drugs, together with 30 potential candidates for application in a number of disease areas that are in different stages of clinical trials (Phase I, II and III) Figure 1 [5]. 

A number of reviews have also detailed the biologically active natural products that are sourced from marine organisms [6,7,8,9,10], highlighting marine organisms as an important resource for the production of new and unique compounds with potential medicinal value. Many of the compounds represented in Figure 1 have been isolated from sponges, ascidians and cyanobacteria. Marine macroalgae are underrepresented in the pharmaceutical pipeline despite the number of biologically active compounds. Marine algae have also been used in a number of other important areas including the food industry [11], agriculture [12] and as a source of third-generation bioplastics [13]. Compounds from marine algae have been shown to exhibit a number of biologically active properties such as anti-microbial [6,7,14], anti-cancer [15], anti-leishmanial [16], anti-inflammatory [17], anti-fouling [18] and anti-protozoal [19] activities. Historically, Australia has been an excellent source of novel marine invertebrate chemistry. Australia has the largest Exclusive Economic Zone (EEZ) on the planet, which is made up of several diverse marine ecoregions. As a source of novel chemistry, Australia’s EEZ has been prolific, with its contribution representing the third largest of newly discovered MNPs during the period 1965–2012, only behind Japan and China, respectively. Of interest is the marine ecoregion of Port Phillip Bay, located on the south eastern coast of Australia in the state of Victoria. During the period 1995–2012, this marine ecoregion has been the dominant source of unreported natural products located within Australia’s EEZ [20]. The primary reason for this being the large amount of habitat variety that is present in this Bay, such as intertidal sandy beaches, mangroves and rocky shores along with tidal habitats like sand beds, seagrass beds and rocky reefs. Port Phillip Bay, located on the southern shore of Victoria (Figure 2), has an area of approximately 2000 square kilometers and an average depth of 13 m. It represents a unique habitat, being shallow enough to be in the photic zone throughout and is known for the cleansing activities of the microphyto- and zoo-benthos and continues to be a region of Australia that yields new species. It is home to approximately 200 different species of macroalgae, a figure which is subject to change due to introduced species that originate from shipping within the Port) and is represented by all three major phyla, namely the brown (Ochrophyta), red (Rhodophyta) and the green algae (Chlorophyta). 

In order to explore the potential of the Port Phillip Bay region as a continuing source for bioactive marine natural products, it is important to review and document the natural products that have been studied and the bioactivities that have been discovered for the marine algae occurring in this region. The methodology adopted to compile this review required that a species list for Port Phillip Bay macroalgae to be created using the Victorian Biodiversity Atlas in conjunction with the Melbourne Museum listings for algae species found within the Port Phillip Bay area [21,22,23]. Table 1 displays the listings of the species of Port Phillip Bay algae that this review focuses upon. A number of these species are not endemic to Port Phillip Bay, and have been sampled worldwide. Thus, this review will not be limited to natural products derived only from marine algae sampled from Port Phillip Bay, but will focus on the global study of natural products from each marine algae species that exists within the Port Phillip Bay region. 

The purpose of this review is to provide a compilation of the natural products found in the marine macroalgae of Port Phillip Bay and discuss the associated trends in their biological activities. Emphasis has been placed on the three major phyla of algae, namely the green (Chlorophyta), the brown (Ochrophyta) and the red algae (Rhodophyta) and the study of their secondary metabolites between 1971 and early 2019. This review provides listings of compounds that are categorized under the following structure classes: terpenoids, sterols/steroids, phenolic acids, phenols, lipids/polyenes, pheromones, xanthophylls and phloroglucinols. Compound classes that include carbohydrates/sugars (polysaccharides, agars and carrageenans), tannins, tannic acids, phlorotannins and fatty acids have been excluded from this review, owing to their ubiquitous nature.

## 2. Chlorophyta (Green Algae)

The phylum Chlorophyta are found to be distinctively green in color due to the presence of chlorophyll a and b occurring in high concentrations. Green algae proliferate within the euphotic zone of the ocean, or where there is sufficient sunlight to perform effective photosynthesis, usually growing within the intertidal zone up to depths of 50 meters. The most common and frequently studied species of green algae found in Port Phillip Bay are within the genera *Caulerpa, Codium* and *Ulva.* Many of the species that comprise the three mentioned genera can also be found in various tropical, sub-tropical and temperate marine climates around the world and are thus not exclusive to Port Phillip Bay. Common types of secondary metabolite classes found within the phylum Chlorophyta include diterpenes, sesquiterpenes, sterols and lipids. This review reports a total of 64 secondary metabolites distributed among 12 species of common green algae of Port Phillip Bay within the period 1971 to early 2019. 

### 2.1. Terpenoids

#### 2.1.1. Diterpenes

Algae from the genus *Caulerpa* would appear to have yielded the majority of diterpene compounds, comprised of both cyclic and acyclic C-20 diterpenes (**1**–**20**) (see Appendix A). Many of the cyclic diterpenoid compounds found within *Caulerpa* show a variety of biological activities. Compound **7**, extracted and characterized in 1985 from *C. brownii*, was shown to exhibit anti-bacterial activity towards *Staphylococcus aureus*, *Bacillus subtilus*, *Escherichia coli* and *Vibrio anguillarum* utilizing the disc diffusion methodology at 100 μg/disc [24]. Metabolites (**5** and **9**) derived from *Caulerpa trifaria* displayed moderate cytotoxic behavior when tested using the brine shrimp assay [25]. Caulerpol (**2**) appears to be found in most species of the genus *Caulerpa*, usually as the dominant constituent of the crude extract. The isolation of caulerpol was of particular importance as it was the first compound with a retinol carbon skeleton isolated from a plant source and was later shown to be easily synthesized from (S)-(−)-α-cyclogeraniol [26]. A number of terpenoid esters (**2a**–**2f**) were also found with R groups representing the fatty acids arachidonic, eicosapentanoic, oleic, linoleic, linolenic and hexadecatrienenoic acids [27]. The acyclic diterpenes of the genus *Caulerpa* exhibit both acetoxy and aldehyde functionalities, much the same as the cyclic diterpenes. A good example being the natural product trifarin (**17**) from *C. trifaria* which contains two acetoxy groups. The usual scenario for these diterpenes is to contain acetoxy groups or both aldehyde and acetoxy groups, but it is rare to observe diterpenes from green algae with two aldehyde functional groups, as seen in compounds **5** (*C. brownii* and *C. trifaria*) and **20** (*C. brownii*) [27]. Diterpenes are also found outside the genus *Caulerpa*, but only within the species *Codium fragile*. Non-polar fractions of *C. fragile* have been shown to yield saturated terpenoid compounds such as trans-phytol (**13**) and its two derivatives, phytyl acetate (**14**) and phytyl palmitate (**15**) [28]. 

#### 2.1.2. Sesquiterpenes

Sesquiterpenes (C-15) are found in a smaller number of *Caulerpa* species as diterpenes but have only been found within the genus *Caulerpa* for the reported green algae of Port Phillip Bay. Sesquiterpenes, with acetoxy and aldehyde functionalities (**21**–**24**), Figure 3, have been shown to be potent feeding deterrents and in some cases cytotoxic towards predatory species of herbivorous fish [29]. It has been suggested that caulerpenyne (**24**), an acyclic acetylenic sesquiterpene found in *C. trifaria*, as a minor secondary metabolite and in *C. taxifolia*, as a major secondary metabolite, is a key player in the cytotoxicity of *Caulerpa* algae towards herbivores [30]. 

This sesquiterpene appears to contribute to the invasiveness of the genus *Caulerpa* by means of its inhibition of the key organic anion transporters, Oatp1d1 and Oct1. These transporters play a role in the toxicity defense of a herbivorous predator of the genus *Caulerpa*, the zebra fish (*Danio rerio*) [30]. Although this particular mechanism of action has only been demonstrated towards zebrafish, it appears to provide some reasoning behind the apparent success of *Caulerpa* as an invasive species of algae [30]. Compound (**23**) was shown to display moderate anti-microbial activity against *B. subtilis* and the marine fungus *Dreschleria haloides*. Furthermore, this compound inhibited the cell division of fertilized sea urchin eggs, demonstrating its antipredatory attributes [31].

#### 2.1.3. Cyclic geranylacetone

All cyclic geranylacetone compounds (**25**–**29**), Figure 4, that have been characterized were isolated from the ethyl acetate fraction of the green alga *Ulva lactuca*. Compounds of a similar structure type, namely, the apocarotenoids, have been studied for their potential germination and growth inhibition qualities [32,33]. Many similar types of C_13_ nor isoprenoids have been found largely in wine, particularly Rieslings. Many of these compounds have been isolated and studied for their floral aromas [34,35]. A distribution of the terpene compounds reported in this review by species and locality is shown in Table 2.

### 2.2. Steroids/Sterols

The sterols (see Appendix A) of *U. lactuca* (**30**–**36**, **40**, **47**) and *U. australis* (**40**, **41**, **42**, **44**–**46**) have been studied extensively [39,40,41,42,43]. Many of the sterols found in the green algae of Port Phillip Bay appear to have a C-19 core skeleton with differing functionalities of the side chain. Of note, are compounds **41** and **42** derived from *U. lactuca*, which exhibit a keto group within the C-19 skeleton. Also of interest are the sterols **38**, **47** and **48**, which all appear to be variants of clerosterol, with **47** and **48** having attained a glycosidic moiety, whereas **38** has a long-chain ester attached. Acetylation of sterol fractions appears to be a tactic employed for easier isolation of these compounds, which was evident in the reported isolation of acetylated codisterol (**37**) from the alga *Codium fragile* [44]. Interestingly, compounds **37**–**39** and **48** were all isolated and characterized from the alga *C. fragile* and following this study, it was suggested that sterol compounds may prove to be useful biomarkers for this genus allowing for easier taxonomic distinction between other members of the Codiaceae family [44,45]. Much of the literature available for the genus *Ulva* involves biological activity studies on polysaccharides [46,47,48,49], many of the sterols found within this genus have shown promising in vitro bioactivity. In particular, the glycosidic sterol (**47**) from *U. lactuca* has been shown to be an effective anti-bacterial, anti-fungal and anti-inflammatory agent in vitro [41]. Compound **46**, displaying an epoxide side chain functionality, appeared to display moderate recombinant aldose reductase inhibition when assayed at 3 μg/mL [39]. This compound outperformed all other sterols in this assay (**41**–**45**) of which, compounds **41**, **42**, **44** and **45** possess a hydroxyl vinyl moiety in place of the epoxide. This suggests that the epoxide side chain moiety must play a significant role in the inhibition shown by compound **46**.

### 2.3. Miscellaneous

Within the reported green algae of Port Phillip Bay, a number of miscellaneous compounds were found that include some lipids (**49**–**52**), bromophenolics (**54**–**58**) and a pigment (**53**), Figure 5. The di-indolo pigment caulerpin (**53**) was found exclusively within algae of the genus *Caulerpa* (*trifaria, brownii, flexilis, racemosa* and *peltata*), perhaps offering a useful chemotaxonomic marker for this genus [50]. The unprecedented bicyclic lipid dictyosphaerin (**49**) was isolated from the endemic Australian alga *Dictyosphaeria sericea.* This appears to be the only natural product reported in the literature for this species and is the only genus of reported Port Phillip Bay green algae outside of *Caulerpa* and *Ulva* that has shown the presence of lipidic compounds. 

It should be noted that this compound was only partially characterized, as neither its relative nor absolute configuration have been described [51]. Compounds **50**–**52** were derived from the petroleum ether fraction of a methanolic extraction of *U. lactuca* along with isofucosterol and some fatty acid compounds [52]. In 1999 a profiling study of Eastern Australian marine algae, which included various green algae (*Caulerpa cactoides*, *C. fragile*, *Codium galeatum*, *Codium lucasii* and *U. lactuca*) [53], resulted in the identification of simple low molecular weight bromophenolic compounds **54**–**58** that were confirmed to be present in varying amounts. A distribution of the sterol, lipids and miscellaneous compounds reported in this review by species and locality is shown in Table 3. A summary of the biological activities is provided in the Appendix A.

## 3. Ochrophyta (Brown Algae)

The Ocrophyta phylum has been the most studied phylum of algae in Port Phillip Bay to date and has yielded the largest number and variety of natural products. *Caulocystis cephalornithos, Dictyota dichotoma* and *Notheia anomala* represent the species of brown algae that have yielded the greatest number of secondary metabolites. Each of these species has shown the presence of an extensive range of phenolic compounds, diterpenoids, sesquiterpenoids and long-chain unsaturated lipids. Reported herein is a total of 281 secondary metabolites isolated from 37 species of brown algae within the period 1971 to early 2019, representing several terpenoid classes, steroids/sterols, lipids and other miscellaneous compound classes.

### 3.1. Terpenoids

#### 3.1.1. Tocotrienols

A part of the vitamin E family, the tocotrienols are a class of terpenoids that are characterized by their unsaturated farnesyl tails attached to a chromane ring, with variations between the types being expressed through substitutions on the aromatic ring or methylation of the hydroxyl group. There are four variations of the tocotrienol (α, β, γ and δ), two of the variants, γ (**59**) and δ (**60**), are reported herein. Two methylated variants of the tocotrienol compound class (**61** and **62**), Figure 6, were also reported here but the identity of compound **62** was not confirmed, as the location of the methyl groups on the chromane ring was unclear. This could mean that the structure of **62** could be either β-tocotrienol or γ-tocotrienol [55]. All tocotrienols were found within the genus *Cystophora*, δ-tocotrienol (**60**) appeared to be the most prolific type being isolated as a secondary metabolite in *Cystophora subfarcinata*, *Cystophora platylobium, Cystophora monilifera*, *Cystophora siliquosa* and *Cystophora retorta* [55,56,57,58]. Both *γ*-tocotrienol and *δ*-tocotrienol have been reported to display a broad range anti-cancer activity including against colon carcinoma and lung cancer [57].

#### 3.1.2. Monoterpenes

Only five monoterpenes (C-10) were isolated from the brown algae listed in this review; in this instance, compounds **63**–**67**, Figure 7, were all derived from the edible alga *Undaria pinnatifida*. All monoterpenes are in the form of loliolide derivatives, differing only in the stereochemistry of a tertiary alcohol and methyl group along with the degree of unsaturation. Loliolide monoterpenes have been well studied for their biological activities [59]. Compound **65**, (+)-epiloliolide, was isolated from the brown alga *Sargassum naozhouense* and showed moderate antioxidant activity scavenging 1, 1-diphenyl-2-picrylhydrazyl (DPPH) free radicals. Further, epiloliolide proved to have anti-microbial properties as well as displaying resistance to the fungus *Candida albicans* and the two bacterial strains *Escherichia coli* and the methicillin-resistant *Staphylococcus aureus* (MRSA) [60]. More recently, loliolide (**67**) was reported to display allelopathic influence on the germination of surrounding plant seeds, which could be a contributor to the relative competitiveness of the brown alga *U. pinnatifida* [61]. 

#### 3.1.3. Prenylated Phenols

Prenylated phenols (**68**–**80**) (see Appendix A) are a significant group of compounds found in brown algae. They are identified from their terpenoid tails, of varying length, which are attached to a phenolic head group which is sometimes further cyclized as in compound **74**. Prenylated phenols were discovered across three different genera and five different species of brown algae. To date, the *Sargassum* genus has yielded the greatest amount of prenylated phenols with compounds **68**–**71** and **77** derived from *Sargassum paradoxum*, and its relative *Sargassum fallax* only boasting two of the same phenols in **68** and **69** [62,63]. Of note was the moderate anti-tumor activity of compound **69** (sargahydroquinoic acid) against P388 (Murine Leukaemia cells) achieving an IC_50_ value of 14 μM, when tested at 1 mg/mL [63]. The dichloromethane (DCM) extract of *S. paradoxum* was assayed against a series of bacteria (*S. aureus*, MRSA and *S. pyogenes*) showing weak to moderate resistance. This was supported by compounds **68**–**71**, isolated from the DCM extract, displaying a similar degree of anti-bacterial activity [62]. Compounds **75** and **76**, isolated from *C. brownii*, are unique in this class and are of particular interest with respect to the molecular phylogeny of the *Cystophora* genus. These compounds display further complexity compared to their counterparts due to the incorporation of a furan ring in their terpenoid tails. Due to this fact, it is theorized that *C. brownii* is perhaps more phylogenically advanced than its *Cystophora* relatives. *C. torulosa*, on the other hand, appears to display only prenylated phenols with lower molecular weight and less relative complexity, such as compounds **78** and **79**, perhaps suggesting that this *Cystophora* species is less phylogenically developed than *C. brownii* [56]. The study of *Perithalia caudata* from the family Sporochnaceae resulted in a number of simple prenylated phenols (**72**–**74** and **80**) being isolated with promising anti-bacterial assays [64,65,66,67]. It should be noted here that it was unclear if **74** was indeed a true natural product of *P. caudata* or perhaps an artefact of the isolation process formed by a ring closure of **72**. Compound **72** was reported to show Minimum Inhibitory Concentrations (MICs) of 3.1 μg/mL for assays against both *C. albicans* and *Cryptococcus neoformans* and an MIC of 6.2 μg/mL against *B. subtilus* [65]. A distribution of the tocotrienols, monoterpenes and prenylated phenol compounds reported in this review by species and locality is shown in Table 4.

#### 3.1.4. Meroditerpenoids

As with prenylated phenols, the meroditerpenoids are primarily found in the genus *Sargassum* with all but compound **81** being natural products of either *S. paradoxum* or *S. fallax*. The meroditerpenoids **82**–**88** and **90**, Figure 8, isolated from *S. paradoxum* all displayed some level of anti-bacterial activity. Compounds **82**, **84**, **85**, **87** and **88** displayed weak anti-bacterial activity against the Gram-positive bacterium *S. pyogenes*, whilst compound **86** outperformed the standard antibiotic ampicillin against the Gram-negative bacterium *P. aeruginosa* [57]. *S. fallax* yielded compounds **86**–**89** and **91**, with compound **87** displaying moderate cytotoxicity towards P388 cancer cells (IC_50_: 17 μM at 1 mg/mL). In contrast compounds **86** and **91** only had IC_50_ values of 32 μM and >27–29 μM, respectively, when measured under the same conditions. Compound **91** (fallachromenoic acid) is also of chemotaxonomic interest, as this is the first chlorinated meroditerpene isolated from *S. fallax* [63].

Interestingly, compounds **89**–**91** appear to all be derivatives of *δ*-tocotrienol with carboxylic acid, aldehyde or halogen functionalities. Compound **81**, technically a meromonoterpene, was isolated from the acetone extract of the plant *Cystophora torulosa* via Sephadex LH-20 size exclusion chromatography. As this is the first meromonoterpenoid isolated from *C. torulosa*, it was suggested that it be a candidate for assessing phylogenic relationships of the *Cystophora* genus [58].

#### 3.1.5. Sesquiterpenes and Monoterpenes

Terpenoids that populate this class are both acyclic and cyclic and are identified by terminal ketone or aldehyde functional groups, followed by C-15 or C-20 terpenoid tails. There are multiple varieties of mono- and sesquiterpenes including farnesylacetone epoxides (**92**–**94**), cyclic farnesylacetones (**102**–**110**), farnesylacetones (**95**–**97**, **101**), geranylacetones (**100**) and geranylgeranal epoxides (**98**, **99**) (see Appendix A). All compounds in this class were isolated and characterised from the alga *Cystophora moniliformis* [58,68,70,71,72]. Taxonomically, this class of compounds is reported to consist of good indicators of the developmental progress of species within the genus *Cystophora* [68]. As these compounds are suspected to be derived from the ubiquitous tocotrienols, the higher abundance of them within *C. moniliformis* provides support to the claim that this alga is the most developed species of *Cystophora*. Many of these compounds have been shown to have weak or no anti-microbial activity, but an anti-tumor assay of a mixture of (**107**) and (**110**) present in a 3:1 ratio, respectively, showed it to possess moderate anti-cancer activity (IC_50_: 45 μM at 1 mg/mL). Furthermore, this same mixture showed moderate anti-fungal ability via the disc diffusion assay against *Trichophyton mentagrophytes* [72]. Crude extracts of *C. moniliformis* have been shown to have quite potent anti-tumor activity against P388 cells, but no single compound has been isolated that appears to account for the high crude extract activity. This supports the idea that the collective effects of monoterpenes and sesquiterpenes are responsible for the anti-tumor activity observed in the crude extract [58,72]. A distribution of the meroditerpene, monoterpene and sesquiterpene compounds that are reported in this review by species and locality is shown in Table 5.

#### 3.1.6. Diterpenoids

The diterpenes (C-20) from brown algae have been extensively studied and reviewed, and none more so than the diterpenes derived from algae within the genus *Dictyota* [73]. The *Dictyota* genus is within the family Dictyotaceae and consists of some 221 species. In this instance, only one diterpene was found outside of the species *D. dichotoma*, a dolastane diterpene from the less studied *D. furcellata* (**143**) [74]. All other diterpenes (**112**–**142**, **144**–**192**) (see Appendix A) reported were from the species *D. dichotoma* which has been extensively studied. A recent review of the diterpenes in question suggested a grouping of diterpenes based on biosynthetic origins and also diterpene cyclization complexity [73]. The grouping separates the diterpenes into three groups, Group I (**112**–**134**, **136**–**140**), Group II (**135**, **141**–**175**) and Group III (**176**–**192**). Group I diterpenes are all compounds derived from the apparent first cyclization of the pre-cursor geranyl-geraniol between positions C-1 and C-10. Group II involves the same cyclization of gernayl-geraniol pre-cursor but between C-1 and C-11, while Group III involves cyclization between C-2 and C-10. Diterpenes of all three groups are reported to exhibit significant anti-tumor, anti-viral and some anti-fouling activities [73]. A distribution of the diterpene compounds reported in this review by species and locality is shown in Table 6.

### 3.2. Steroids/Sterols

Brown algae are responsible for the production of several sterol compounds, but when compared with green algae, there are some differences. Firstly, green algae produce a greater variety of sterolic compounds, and secondly, some structural differences are apparent. Green algal derived sterols appear to have a higher inclination towards glycosidic moieties along with long lipid esters that branch from the hydroxyl group on ring A of the steroid skeleton (**38**, **47**, **48**). Steroids derived from the brown algae appear to have no glycosidic attachments but display more diversity in the lipidic chains sprouting from the D ring (**200**–**202**). Furthermore, the brown algae *C. brownii* produces two new sterols (**201**, **202**) which was of particular interest, as prior to this there had only been two other occasions where polyoxygenated sterols had been isolated from brown algae. The steroid fucosterol (**200**) has been found in a number of brown algae *S. linearfolium*, *C. sinuosa*, *D. dichotoma* and *C. spongiosum* [43,75,76]. This common steroid is of interest as it has been reported to be a potent acetylcholinesterase (AChE) inhibitor for symptomatic treatment of Alzheimer’s disease [77]. Furthermore, fucosterol (**200**) has been shown to be a potent anti-malarial agent displaying an IC_50_ value of 7.48 μg/mL [78]. Other common sterols such as cholesterol (**199**), desmosterol (**196**) and campesterol (**198**) were also found within *C. sinuosa*, *Cladostephus spongiosus* and *C. brownii*. Also found among the same algae including the prolific *D. dichotoma* are the relatively common steroids Brassicasterol (**193**), dehydrocholesterol (**194**), Poriferasterol (**195**) and Clionasterol (**197**). A summary of the isolated steroids/sterols is given in the Appendix A. 

### 3.3. Lipids

Brown algae are known to produce large amounts of straight-chain fatty acids and saturated or polyunsaturated lipids. Lipidic compounds were found to be highly abundant in the brown alga *C. cephalornithos*, where they were present as saturated fats (**210**–**214**), unsaturated fats (**215**–**219**), straight-chain ketones (**203**–**209**, **222**) and diketones (**223**) as well as secondary alcohols (**220**, **221**) [79] (see Appendix A). A study that yielded the lipids reported herein from the brown alga *C. cephalornithos* showed large variance in relative amounts of lipids based on collection site and season. For example, a larger amount of the alkene **215** was present when the alga was collected during September from sites around Tasmania (Spring). This was suggested to be due to an increase in alkene production during rapid growth of the alga or potentially due to smaller alkene losses in the Southern hemisphere winter [79]. The brown alga *Lobophora variegata,* commonly found in the Canary Islands, was also shown to yield saturated and unsaturated ketones (**225**–**227**) [80]. This particular species of algae has been reported to suffer extremely low levels of microbial infection and has also been studied in an ecological perspective with particular interest in its role in absorbing heavy metal ions [81]. A 2015 study examined the anti-bacterial properties of the compound lobophorone E (**227**) against both Gram-negative and Gram-positive bacteria, but it was shown to be significantly outperformed by the positive control ciprofloxacin [80].

#### 3.3.1. Polyenes

The polyenes detailed in this review where distributed in the following algae: *N. anomala* (**228**–**231**) [82], *C. torulosa* (**228**, **231**) [55] and *C. retorta* (**228**, **231**) [58] (see Appendix A). It appears that the genus *Cystophora* exhibited the more highly unsaturated polyenes, whereas the polyenes **229** and **230** were only found in the alga *N. anomala*. Compounds **228** and **231** were reported to be potent lipoxygenase inhibitors, with this type of inhibition being important in the prevention of psoriasis, asthma, rhinitis and arthritis, with IC_50_ values of 40 μM and 5.0 μM, respectively [83].

#### 3.3.2. Oxy/Epoxy lipids

All Oxylipids and Epoxylipids were found in the brown alga *N. anomala* (see Appendix A). Of interest are the Oxylipids **232** and **233** [84], which have been targets of synthetic studies due to their interesting 2, 5-disubstituted-3-oxygenated tetrahydrofuranyl motif [85] that appears to be responsible for the biological activity of both compounds. Compounds **232** and **233** have both shown potent nematocidal activity against the parasitic nematode species *Trichostrongylus colubriformis* and *Haemonchus contortus* [84]. A possible biosynthetic pathway to Oxylipids (**232**, **233**, **246**–**254**) was suggested via C-18, C-20 and C-22 lipidic pre-cursors, showing a possible link to the epoxylipid structure also found in *N. anomala* (**234**–**245**) [86]. A distribution of the steroids, lipids, polyenes and oxy/epoxy lipids compounds reported in this review by species and locality is shown in Table 7.

### 3.4. Phenols

#### 3.4.1. Phloroglucinols

Phloroglucinols appear to be one of the most widely spread classes of secondary metabolite within the phylum Ochrophyta when considering the 13 species of brown algae that show the presence of phloroglucinols. Although widely spread at the species level, this compound class was only found among two genera of algae, namely, *Cystophora* and *Zonaria*. It has been stated that the phloroglucinols are of taxonomic importance to *Cystophora* algae providing an alternate means of tracking evolutionary development of species within this genus [58]. However, this method of tracking phylogeny has shown apparent deviations to the current theory of species evolution within this genus [90]. The species *C. subfarcinata* (**255**, **256**, **259**–**262**, **265**, **269**–**271**), *C. monilifera* (**255**, **259**, **261**, **262**, **265**, **266**, **269**, **270**, **272**), *C. retroflexa* (**255**, **262**, **263**, **265**, **269**–**271**) and *Z. spiralis* (**257**, **258**, **264**, **273**–**275**) (see Appendix A) represent the brown algae that have produced the greatest number of phloroglucinol compounds [57,58,91,92]. An interesting trend appears within the species *Z. spiralis*, where it is observed that rather than yielding primarily monocyclic phloroglucinols, as in the genus *Cystophora*, *Z. spiralis* appears to mainly produce bicyclic derivatives expressed as hemiketals and chromones [93]. Other members of the *Zonaria* genus, including *Z. turneriana*, *Z. crenata* and *Z. angustata* which also show monocyclic phloroglucinols as the major secondary metabolites [94]. Hemiketals and chromones (**273**–**275**) from *Z. spiralis* have shown inhibitory activity against prominent neurodegenerative disease kinase targets and also anti-bacterial activity (**257**, **258**, **273**, **275**) against *B. subtilus*, with all compounds having IC_50_ values between 2.5 and 10.0 μM [93]. Other monocyclic phloroglucinols from *C. subfarcinata* (**252** and **267**) and *C. monilifera* (**255**, **262**, **266** and **270**) have displayed weak anti-bacterial activity against the Gram-positive bacteria *Streptococcus pyogenes*, only showing minimal inhibition zones when tested using the disc diffusion assay at 1 mg/mL. Compound **256** showed equal activity against Gram-positive and Gram-negative bacteria, *S. aureus* and *P. aeruginosa*, respectively, but once again, all compounds from *C. monilifera* were substantially outperformed by the standard antibiotic ampicillin [57]. Compounds **265** and **268** both found within *Cystophora* and *Zonaria* have each shown moderate anti-bacterial activity against *S. aureus* and *B. subtilis* [95].

#### 3.4.2. Phenols/Phenolic acids/Resorcinols

Phenolic compounds are found throughout brown algae, primarily in the form of a phenolic, phenolic acid or a benzopyranone head group attached to a lipidic tail. These types of phenols have been found across four genera including the prolific *Cystophora* and *Sargassum* as well as *Caulocystis*, *Colpomenia* and *Lobophora*. The benzopyranones (**276**, **277**) were found in this instance only within the species *C. cephalornithos*. These particular compounds, found also in the plant *Ginkgo biloba L*., have been shown to be biological derivatives of ginkgolic acids, which are themselves a form of the anti-inflammatory agent salicylic acid [96]. In contrast to salicylic acid, these compounds, in the form of ginkgolic acid or benzopyranone, are responsible for allergic contact dermatitis (ACD) [96]. *C. cephalornithos* also yielded several resorcinols (**285** and **286**), phenolic acids (**278**–**282**), phenols (**283** and **284**) and dihydroxy phenolic acids (**288, 289**). Compounds similar to **285**–**287** and **290, 291** have been studied previously for their anti-cancer properties. In particular they have been found to have strong activity against human colon cancer cells (HCT-116 and HT-29) [97]. The resorcinol **286** was shown to have the highest cytotoxicity with IC_50_ values of 31.45, 35.27 and 24.28 μg/mL against SMMC7721, K562 and HeLa, respectively [98]. This class of lipidic resorcinol has also been shown to exhibit various anti-tuberculosis activities [99]. Polar extracts of the alga *C. peregrina* yielded a number of common low molecular weight aromatic acids such as compounds **292**–**295**. These low molecular weight compounds were identified on the basis of GC–MS experiments [100]. Compounds **296** and **297**, isolated from the brown alga *L. variegata*, were shown to exhibit small to moderate inhibition against the Gram-positive bacteria *S. aureus* [80]. A summary of the phenols/phenolic acids/resorcinols is given in the Appendix A. A distribution of the phloroglucinols, benzopyranones, phenolic acids, phenols and resorcinol compounds reported in this review by species and locality is shown in Table 8.

### 3.5. Miscellaneous

Several other secondary metabolite classes have been isolated and characterized from the brown algae considered in this review. Classes include 1-deoxysphingoid bases, pheromones, bromophenolics and xanthophylls. The 1-deoxysphingoid base **298** (3-epi-xestoaminol C) was isolated from the brown algae *Xiphophora chondrophylla* and was the first 1-deoxysphingoid that had been isolated from brown algae [101]. Compound **298** had its absolute configuration determined using the Mosher method and was subsequently reported to have quite remarkable multifaceted bioactivity [101]. Initial studies demonstrated antitubercular activity with an IC_50_ value of 19.4 μM with inhibition against *Myobacterium tuberculosis* (H37Ra). This was followed by confirmation of growth inhibition against both *S. aureus* and *S. cerevisiae* with IC_50_ values 17.0 μM and 17.1 μM, respectively. 3-epi-xestoaminol C showed great promise when assayed against human leukemia cells (HL-60) achieving an IC_50_ of 8.8 μM, which was followed by an IC_50_ of 18.0 μM when assayed against human embryonic kidney cells (HEK) [101]. 

Brown algae have long been known to possess a variety of pheromone compounds with a number of functions contributing to the reproductive cycle, which have been studied extensively across a number of species. A study on the species *Dictyopteris acrostichoides* showed the largest number of C-11 pheromones (**299**–**305**, **313**, **316** and **317**). These compounds were isolated from extracts of female gametes of *D. acrostichoides* [102]. Many of these compounds are produced by the organism with the primary function of attracting male gametes to complete sexual reproduction, but some have also been shown to function as effective anti-predation agents, or may even be used to interfere with other pheromone communication systems of competing alga [103,104]. Many species of brown algae reported herein, including *X. chondrophylla*, *Scytosiphon lomentaria* and *Hormosira banksii*, have been found to produce the sexual pheromone hormosirene (**309**), suggesting that this is a particularly important pheromone for the phylum Ochrophyta [103,105]. The brown alga *C. peregrina* has been found to exhibit the pheromone (**304**), which has long been suspected as a sperm attractant for this and many other species [106]. A mixture of miscellaneous pheromone compounds has also been found distributed across seven species of brown algae including *D. acrostichoides* (**306**, **313**), *X. chondrophylla* (**307**), *S. lomentaria* (**307**), *H. banksii* (**308**), *P. caudata* (**310**–**312**), *C. spongiosus* (**313**, **315**), *Macrocystis pyrifera* (**314**) and *U. pinnatifida* (**314**).

The simple low molecular weight bromophenolic compounds **318**–**322** were found to be present in varying amounts in many Australian algae including *C. spongiosus*, *C. sinuosa*, *E. radiata*, *Homoeostrichus sinclairii*, *H. banksii* and *Phyllospora comosa*, and this was confirmed by a bromophenolic distribution study of many brown, red and green algae [54]. Interestingly, a rare bromophenolic of the C-6 C-4 C-6 arrangement was isolated from the brown alga *Colpomenia sinuosa* (**323**) via a bio-activity directed isolation. This metabolite was found to be responsible for the cytotoxicity that the crude extracts of this alga displayed [107].

Xanthophyll compounds such as fucoxanthin (**329**) have been found consistently throughout brown algae and have been the topic of a number of review articles due to their potential anti-cancer/anti-tumor applications [108,109]. Fucoxanthin has been found in a number of brown algae species (*U. pinnatifida*, *S. lomentaria*, *C. spongiosus*, *Halopteris pseudospicata*, *Sargassum vestitum*) and has, among some studies, largely contributed to the anti-cancer activity of crude extracts [109]. Due to the significant activity and interest in fucoxanthin (**329**), some work has been undertaken to explore the metabolism of this compound when consumed via edible brown algal species such as *U. pinnatifida*. The compounds fucoxanthinol (**328**) and amarouciaxanthin A (**327**) are the natural metabolites of fucoxanthin (**329**) and are thought to play a key role in the anti-cancer activity that has been associated with diets high in fucoxanthin containing algae [108,110]. A number of studies have also described the isolation of apo-carotenoids from the brown algae *S. lomentaria* (**324**–**326**), *C. spongiosus* (**325**, **326**) and *Z. spiralis* (**326**). These apo-carotenoids are known to be oxidized derivatives of fucoxanthin (**329**) and appear to display some feeding deterrent activity (**325**, **326**) [69,93,111].

Non-polar extracts of the brown alga *A. paniculata* were found to be a rich source of the furanic esters **330** and **331**. These compounds are closely related to the furan fatty acids obtained from the sap of the rubber tree *Hevea brasiliensis* that plays a major role in the fabrication of latex [112]. The common low molecular weight compounds picolinic acid (**332**) and trimethylamine (**333**) were identified in trace quantities in the brown alga *C. peregrina* through use of GC–MS [100]. 

An interesting polyketide macrolide was isolated from the brown alga *L. variegata*, with the compound lobophorolide (**334**) being isolated as 1.2 × 10^−4^% of the algal dry mass. Lobophorolide was assayed for both its anti-fungal and anti-tumor properties and found to exhibit good potency in both assays [113]. Although this study noted that due to the shared structural motifs of lobophorolide (**334**) with that of bacterial natural products, it was possible that this compound was derived from a symbiont of *L. variegata* which was further substantiated by the relatively low isolation yield of the natural product. In a 2015 study, several polyketides (**335**–**339**) were isolated from *L. variegata* of which only **335** displayed any notable biological activity. When assayed against *S. aureus*, compound **335** displayed significant growth inhibition, but was relatively ineffective against both *E. coli* and *E. faecalis* [80]. A summary of the miscellaneous compounds isolated from brown algae is given in the Appendix A. A distribution of the miscellaneous compounds reported in this review by species and locality is shown in Table 9. A summary of the biological activities for compounds isolated from brown algae is given in the Appendix A.

## 4. Rhodophyta (Red Algae) of Port Phillip Bay

The red algae of Port Phillip Bay are the least studied phylum of algae, but appear to display the most diverse chemistry, much of which has been reported to display significant biological activity. In particular, the genera *Laurencia* and *Plocamium* have been the source of many of the natural products. The red algae of Port Phillip Bay appear to display a number of terpenoid like compounds, but unlike brown or green algae, many of the red algae contain highly halogenated terpenoids. This appears to set them apart in terms of both structural diversity and biological activity. This review documents 163 natural products that have been derived from 22 species of red algae.

### 4.1. Terpenoids

#### 4.1.1. Halogenated Monoterpenes

The halogenated monoterpenes of the phylum Rhodophyta have been found distributed across the species *Plocamium angustum* (**348**–**353**), *Plocamium mertensii* (**340**–**347**), *Plocamium costatum* (**351**, **354**–**364**) and *Plocamium leptophyllum* (**365**). *P. mertensii* was first reported to contain halogenated monoterpenes in a 1977 study where compound **346** was reported to be a major metabolite of this species [117]. This class of secondary metabolite has been notoriously difficult to secure the correct structures for, primarily due to the high number of halogenated substituents. As a result, a number of previously identified secondary metabolites have had structure re-assignments, many of which were reported in a study of *P. mertensii* [118]. The natural product originally assigned structure **346** was subsequently corrected to **340** (mertensene), using both on-line and off-line methodologies to achieve unequivocal structure characterization. This compound, together with compound **341**, has previously shown insecticidal and growth inhibition against some insect species [119]. Compounds **340**–**342** were all shown to be effective antifeedant agents against a range of pest insect species, all of which displayed some toxicity towards at least one species of insect [120]. Compound **342** showed moderate antitubercular activity and cytotoxicity as well as high potency anti-algal activity toward the alga *Chlorella fusca* [121]. It should also be noted that 3:1 methanol:dichloromethane crude extracts of the red alga *P. mertensii* displayed anti-tumor, anti-viral and anti-fungal activities [118]. In a more recent study of the red alga *P. angustum*, the compounds plocamenone (**352**) and isoplocamenone (**353**) were isolated and characterized. Plocamenone was subsequently tested for cytotoxicity against P388 tumour cells showing promising IC_50_ values of 157.5 ng/mL and >97.5 ng/mL when derived from two separate samples. This same study assayed a mixture of plocamenone and isoplocamenone which achieved an IC_50_ value of >97.5 ng/mL, but isoplocamenone was unable to be assayed individually due to its instability [122]. The alga *P. costatum* was studied in a phytochemical capacity as early as 1976 when two separate papers were published wherein both independently reported the identification of halogenated monoterpenes [123,124]. The non-polar extracts of *P. costatum* yielded the natural products costatone (**351**), costatolide (**354**) and the acyclic costatol (**355**) which was achieved by means of conventional isolation and characterization via single crystal X-ray diffraction [123,124]. In 2014 *P. costatum* was revisited and studied using HPLC–UV–MS–SPE–NMR analysis yielding, among others, the natural products **357**, **358**, **361**–**364**. All compounds from this study underwent anti-microbial screening against *C. albicans*, *M. smegmatis*, *S. aureus* and *E. coli*, but were all found to be inactive [125]. The compound aplysiaterpenoid A (**365**) was isolated from the red alga *P. leptophyllum* in a bioassay guided fractionation using antifeedant activity as the guiding factor in isolation. In this situation aplysiaterpenoid A (**365**) demonstrated potent antifeedant activity against a number of gastropods and other herbivores, achieving complete inhibition with only 40 µg of compound. Inhibited herbivorous species included the gastropods; *Omphalius pfeifferi* and *Turbo cornutus*, the abalone *Haliotis discus* and the sea urchin *Strongylocentrotus intermedius* [126]. A summary of the halogenated monoterpenes isolated from red algae is given in Appendix A.

#### 4.1.2. Parguerenes

Investigations of the parguerenes as secondary metabolites have revealed the presence of at least two substructure classes; deoxyparguerenes (**366**–**371**) and parguerenes (**372**, **373**) together with the isolation of a potential biosynthetic intermediate to the parguerenes (**374**) [127,128]. All compounds reported herein have been isolated from the red alga *Laurencia filiformis*, but this compound class has also been found in the sea hare *Aplysia dactylomela*. The parguerenes have been reported to have highly cytotoxic properties, and as a class have been studied on a structure activity basis. Studies such as these have shown that the cytotoxicity of parguerene compounds is dependent upon the presence of acetoxy groups at the C-2 position and bromine at the C-15 position. This is evident in the high growth inhibition activity of **367** against P388 and HeLa cell types, achieving IC_50_ values of 8.5 and 6.3 μg/mL, respectively [129]. In a separate study, this compound was also shown to act as a potent feeding deterrent against the abalone *Haliotis discus hannai* and the young sea urchins *Strongylocentrotus nudus* and *Strongylocentrotus intermedius* [130]. Compound **366**, which contains the C-2 acetoxy and C-15 bromine motif, also showed moderate cytotoxicity towards Ehrlich carcinoma. More impressive though, was this compound’s ability to act as an anthelmintic agent, taking only 30 mins at a concentration of 10% w/v to achieve paralysis of the worm species *Allolobophora caliginosa*, whereas the standard drug mebendazole takes 4 hours to achieve the same result [131]. Furthermore, both compounds **366** and **367** have been studied for their ability to act as P-glycoprotein inhibitors, indicating potentially significant applications in chemotherapeutic treatment, specifically against multidrug resistant cancers [128]. A summary of the parguerenes isolated from red algae is given in Appendix A.

#### 4.1.3. Chamigrenes

Algae of the genus *Laurencia* have been known to produce chamigrene-type compounds since the 1970s [132]. The two species that have been shown to produce a number of these chamigrene-type compounds are *Laurencia filiformis* (**377**, **379**–**385**) and *Laurencia elata* (**375**–**378**, **381**) (see Appendix A). The chamigrenes differ from other red algae derived terpenoids as they are both polyhalogenated and contain spiro centres, providing a challenge for structure elucidation, as was the case for pacifenol (**377**) [133]. Chamigrenes, as a class of compounds, have been reported to express anthelmintic behavior and some cytotoxic properties have also been demonstrated [134,135]. By far the most studied chamigrenes appear to be elatol (**378**) and pacifenol (**377**). Elatol exhibits a chloro vinyl moiety and is the major constituent of the red alga *L. elata* (now reclassified as *Corynecladia elata* see Appendix A) [132]. This compound exhibits potent cytotoxicity against HeLa and Hep-2 cells (IC_50_ 1.3 μM and 2.0 μM, respectively) [134]. Elatol also appeared to have moderate to high antibiofouling properties inhibiting the seaweed pathogens *Alteromonas* sp1., *Alteromonas* sp2., *Proteus mirabilis*, *Proteus* sp., *Cytophaga-Flavobacterium*, *Vibrio* sp. and also showing mild inhibition towards the human pathogen *S. aureus* [136]. An interesting 2017 study also indicated that elatol (**378**) appears to play a large role in the predation of red algae of the genus *Laurencia* by the sea hares *Aplysia* with theories that this compound appears to be a useful foraging cue for *Aplysia* [137]. Compounds **379**, **380** and **381** have shown moderate activity in the brine shrimp (*Artemia salina*) bioassay, but the strongest activity was observed for pacifenol (**377**) where 90% mortality was observed at a concentration of 23 μg/mL after 24 h [138]. A distribution of the Halogenated monoterpenes, parguerenes and chamigrene compounds reported in this review by species and locality is shown in Table 10.

#### 4.1.4. Laurenes

Laurene (**389**) and its structurally related derivatives (see Appendix A) are known to be prevalent throughout the genus *Laurencia*. These compounds are also thought to be the source of sesquiterpenes found within the sea hares of the genus *Aplysia* as they are frequently grazing on *Laurencia* algae [144]. All laurenes reported herein have been derived from the red alga *L. filiformis* (**386**–**397**) and are all variations of compound **389** (laurene) which is found throughout this genus. Biosynthesis of these compounds has been postulated and discussed [145]. Compounds **386**, **390**, **393** and **397** showed cytotoxicity toward P388 cancer cells with IC_50_ values ranging from >34–43 μM when tested at a concentration of 1 mg/mL, although it should be noted that compounds **390** and **397** were unstable and degraded during the course of the assays [146]. Compounds **387** and **390** were assayed for their anti-cancer activity and whilst compound **390** exhibited strong cytotoxicity against the cancer cell line NSCLC-N6 (IC_50_ 26.5 μM), compound **387** was found to possess only weak cytotoxicity [147]. It was also found that compound **386** appeared to have significant anti-bacterial activity against methicillin-resistant *Staphylococcus aureus* (2 × MIC of 6.25 μg/mL) and moderate activity against vancomycin (VCM)-susceptible *Enterococcus faecium* [148]. In a separate study, this particular compound exhibited inhibition of *Mycobacterium tuberculosis* [149].

#### 4.1.5. Sesquiterpenes

All sesquiterpenes were isolated from the genus *Laurencia*, Figure 9. Heterocladol (**400**), isolated from *L. filiformis* collected in both South Australia [145] and Victoria [146], first had its absolute configuration determined in 1977 via crystallographic methods [145]. This was followed by the discovery of austradiol acetate (**398**) and austradiol diacetate (**399**) in a separate study in 1982, thereby cementing eudesmane sesquiterpenes as a prominent secondary metabolite of *L. filiformis* [150]. The compound aplysistatin (**401**) was isolated from *L. filiformis* in 1981, along with hydroxyaplysistatin (**402**) [151]. 

Aplysistatin (**401**) was subjected to a biological activity assessment in subsequent studies and displayed anti-malarial, anti-inflammatory and selective enzymatic suppression activities, the results of which were summarized in recent reviews [152,153,154]. As a result of both the interesting biological activity of aplysistatin and also its relatively unique oxepane ring system, it has been extensively studied in order to achieve a stereo selective synthesis [155]. The compound elatenyne was first isolated and characterized from the non-polar extracts of the red alga *L. elata* [156], and initially assigned the structure **403** [156], but was re-assigned the structure **404** in a later study [133]. In another study of the same compound Gage-Including Atomic Orbital (GIAO) modelling calculations were used on ^13^C NMR spectra in an attempt to solve the relative stereochemistry of this compound but was only able to narrow down the relative stereochemistry to a small number of diastereoisomers [157]. Elatenyne was further studied using the crystalline sponge methodology in 2016 as the absolute configuration of **404** still remained unknown, and this method provided the unequivocal absolute configuration of Elatenyne (**404**) [158]. Synthetic approaches to producing elatenyne were also investigated and yielded a pair of diastereomers of elatenyne [159]. 

#### 4.1.6. Lauroxocanes (C_15_ acetogenins)

In the context of Port Phillip Bay marine algae, lauroxocane compounds have been found in the red algae *L. filiformis* (**405**, **406**) and *L. elata* (**407**, **408**), Figure 10 [133,150]. 

Lauroxocane type compounds have been the target of synthetic studies due to their complex stereochemical nature. As a result, a viable synthetic pathway was established for these compounds in 2012 [160]. A number of compounds from the lauroxocane class, some of which are isomers of **407** and **408**, have been isolated from *Laurencia obtusa* and have been shown to have insecticidal activity against the ant species *Pheidole pallidula* [161]. The lauroxocane **408** was tested for anti-cancer activity and showed no appreciable activity [133], which was in accordance with many of the other lauroxocanes that have been found to have poor cytotoxicity [162]. The compound *cis*-dihydrorhodophytin (**405**) along with other lauroxocanes isolated from the sea hare *Aplysia brasiliana* displayed antifeedant activity. This compound was applied to small beetle larvae and offered to swordtail fish (*Xiphophorus helleri*) along with controls and it was observed that the beetle larvae with **405** applied were usually outright rejected by the fish, whereas the controls were consumed without hesitation [163]. A distribution of the laurene, sesquiterpenes and lauroxocane compounds reported in this review by species and locality is shown in Table 11.

#### 4.1.7. Polyhalogenated Indoles

All polyhalogenated indole compounds (see Appendix A) reported in this review were isolated from *Rhodophyllis membranacea* (**409**–**424**), a red alga sampled from Moa Point [165] and also Seal Reef [166], New Zealand. Isolation of the bromochloroiodoindoles **410**–**412** and **417** is of note, as they contain three types of halogen which is observed very rarely in marine natural products derived from algae. Compounds **409**, **413**, **415**, **410**, **421** and **424** were assayed against the HL-60 cell line and were found to have anti-cancer activity displaying IC_50_ values of 38, 78, 61, 49, 28 and 61 μM, respectively. It was determined, via re-extraction, that compound **422** was isolated as an artefact of compound **424**. This artefact is believed to have occurred through aldol condensation of the keto moiety of **424** with acetone that was used during purification. This was verified after a re-extraction of *R. membranacea* was performed in the absence of acetone. Compounds **409**, **413**, **415**, **410**, **421** and **424** were all found to have anti-fungal activity, compound **421** showed activity (IC_50_ 23 μM) comparable to the standard cycloheximide.

#### 4.1.8. Polyhalogenated Hydrocarbons

This class of secondary metabolites has been reported to be particularly difficult to characterize due to the number of substituted heteroatoms. A large amount of substituted bromines, non-aromatic double bonds, hydroxyl groups and acetoxy functionalities make this class of compound distinct, but also challenging to determine in terms of absolute structures. These compounds have been reported in only two species within this review, *Ptilonia australasica* (**425**–**430**, **432**, **433**) [167,168] and *Delisea pulchra* (**427**, **431**, **434**) [169,170] (see Appendix A). Most compounds reported here were studied for their anti-microbial properties. Compound **427** only displayed moderate to low activity against Gram-positive bacteria (*M. luteus*) inhibiting at 5 μg. Similarly, compounds **431** and **434** also showed low to moderate activity against the Gram-positive Bacteria *M. luteus* but was also able to inhibit the growth of Gram-negative bacteria (*E. coli*). All three compounds displayed moderate anti-fungal properties against the fungus *Puccinia oxalis* [171]. Interestingly, compound **431** also showed the ability to moderately inhibit the enzyme tyrosine kinase with a % Residual Enzyme Activity (REA) of 31.7% being achieved at a concentration of 200 μg/mL [171]. Compound **425** isolated from the alga *P. australasica* was assayed against PC3 cells where it demonstrated some promising anti-cancer activity achieving an IC_50_ value of 0.44 μM. This compared favorably to the positive control, doxorubicin, which was reported to have an IC_50_ value of 0.360 μM [168].

#### 4.1.9. Halogenated Furanones

The red alga *Delisea pulchra* (**435**–**466**) has been a prolific source of halogenated furanones and this species produces large amounts of this class of secondary metabolite. *D. pulchra* has been sampled from the New South Wales Coast and as far south as Palmer station on the Antarctic Coast [171,172]. Samples of *D. pulchra* were found to contain large amounts of halogenated furanone compounds when sampled from both locations. The furanones isolated vary in structure primarily by locality and the type of halogen substitutions present. Regarding sampling locality, algae sampled from the Antarctic coast (Palmer station) appear to exhibit different halogenated furanones to those sampled from the east coast of Australia. For example, the novel compounds known as the pulchralides A–C **463**–**465** were found to be present, along with monocyclic furanones, in samples obtained from the Antarctic Coast [172]. This is in contrast to samples from the Australian coast where normally only the monocyclic furanones containing the same type of lactone ring substituted with halogens are found [169,170]. All compounds in this class have exhibited low to moderate anti-microbial activity. Compounds **441** and **436** were both found to be very active against the bacterial strains *E. coli*, *M. luteus* and *B. subtilis* at a concentration of 1 μg [171]. Compounds **441** and **436**, were also tested for their anti-fungal properties against *P. oxalis* where they displayed inhibitions of 26 mm and 31 mm, respectively when tested at 5 μg [171]. A structure-activity relationship study of the reported halogenated furanones has been performed against an array of cancer cell lines. This study showed that compounds containing an exo-cyclic double bond substitution on the lactone moiety, as is displayed in compounds **435**, **436**, **440** and **441**, appear to be more active than those without it (**452**–**462**) [171]. A large number of halogenated furanones were also tested for antiplasmodial activity against *P. falciparum* clones in vitro, and compounds **445** and **446** had notable IC_50_ values of 2.8 and 2.2 µg/mL, respectively [171]. It was apparent in these assays that many of the halogenated furanones that were tested displayed little or no antiplasmodial activity. For those that did demonstrate small amounts of activity, the pattern appeared to follow that of the anti-cancer assays performed, where the active compounds have both an exocyclic double bond adjoining the lactone ring and also either a hydroxyl or acetyl functionality. A summary of the halogenated furanones isolated from red algae has been provided in the Appendix A. 

### 4.2. Steroids

All sterol compounds reported here were found in the red alga *A. armata* (**467**–**473**) (see Appendix A). The sterol composition of this alga was studied quantitatively, and it was found that the major sterol constituent was cholesterol (**471**) [43]. This agrees with the conventional idea that cholesterol is generally present in algae of the phylum Rhodophyta as the major constituent of sterol extracts [173]. Sterol constituents for this alga were extracted using dichloromethane and saponified using potassium hydroxide and ethanol with a period of reflux in diethyl ether. Varieties of steroids found in *A. armata* do not differ greatly from the steroid classes found in the algae of Chlorophyta and Ochrophyta, but simply differ in the number of steroids that have been reporte [43]. A distribution of the Polyhalogenated indoles, polyhalogenated hydrocarbons, polyhalogenated furanones and steroid compounds reported in this review by species and locality is shown in Table 12.

### 4.3. Miscellaneous

Simple bromophenolics were found to be present in a number of red algae species including *C. officinale* (**474**–**478**), *P. lucida* (**474**–**478**), *C. secundatus* (**474**–**478**), *A. anceps* (**474**–**478**), *J. sagittata* (**474**–**478**), *D. pulchra* (**474**–**478**) and *S. robusta* (**474**–**478**) and *P. angustum* (**475**–**478**) [54]. This was to be expected with the large number of higher molecular weight brominated terpenoids and laurenes that populate the algae of this phylum. As with brown algae, a number of red algae exhibited the presence of xanthophyllic compounds such as fucoxanthin (**479**) and β, β-Carotene (**484**). These have been studied in detail for their anti-cancer properties [108,109].

Xanthophyll compounds such as Fucoxanthin (**479**) and zeaxanthin (**482**) were isolated in abundance from a number of red algae such as *L. botryoides* (**480**, **483**, **484**), *M. abscissa* (**480**, **484**, **496**, **498**, **499**), *A. ciliolatum* (**481**, **482**, **484**), *C. clavulatum* (**481**, **482**, **484**, **496**, **498**) and *P. capillacea* (**481**, **482**, **484**, **496**).

Macrocyclic γ-Pyrones (**486**–**488**) were obtained from the red alga *Phacelocarpus peperocarpos* collected in South Australia. This has been the only instance of secondary metabolite isolation from this species, but many other varieties of γ- and α-pyrones have also been found within the alga *Phacelocarpos labillardieri*, which is also considered to be synonymous with *P. peperocarpos* [178,179,180]. It has also been suggested that biosynthesis of these compounds could occur through a pathway that utilizes a linear diketo acid [179]. To date, macrocyclic enol pyrones of this type have not been found in other natural sources, suggesting that this could potentially prove an important marker for this genus of red algae. γ-Pyrones have also been found within the species *Ptilonia australasica*, with compounds (**490**–**492**) representing the only halogenated γ-Pyrones reported in this review [167,168]. These compounds were found to be more prevalent in the non-polar extracts of *P. australasica.* Compound **490** was assayed against human prostate adenocarcinoma (PC3) cells displaying an IC_50_ value of 10.0 μM, however this was outperformed by the positive control compounds of taxol and doxorubicin that achieved IC_50_ values of 0.002 μM and 0.360 μM, respectively [168].

The red alga *L. filiformis* was shown to have two miscellaneous metabolites, an aromadendrene (**485**) and a lipid with an aldehyde functionality (**489**). A cyclic lipid was found to present in *C. clavulatum* (**497**). The alga *P. costatum* was also shown to have a linear diterpene compound (**500**). *G. filicina,* a red alga from the coast of Japan (now reclassified as *G. subpectinata* see Appendix A), was shown to contain both pyrogallol compounds (**493**, **494**) and a cyclic ketone (**495**) [181,182] which were examined for their biological activity. In a 2012 study a methylated derivative of compound **493** was isolated from the pacific oyster *Crassostrea gigas*. This compound was shown to be an active antioxidant agent displaying potent activity in DPPH assays [183,184]. A summary of all the miscellaneous classes of compounds isolated from red algae is given in the Appendix A. A distribution of the miscellaneous compounds reported in this review by species and locality is shown in Table 13. A summary of the biological activities for compounds isolated from red algae is given in Appendix A.

## 5. Conclusions

This review describes the distribution of 508 natural products derived from algae that can be found within the Port Phillip Bay region in Victoria, Australia. Of the 193 species of algae that are commonly found within Port Phillip Bay, 71 species have been studied and documented for phytochemical purposes and have yielded an array of natural products. Figure 11 displays the distribution of these natural products among phyla and from how many species they were derived from. Figure 12, Figure 13 and Figure 14 show the distribution of compound class amongst the species discussed in this review.

Brown algae have shown the largest number of natural products along with the largest number of species that have been studied for biological activity, whether by crude extract or pure compound evaluation. Green algae and red algae appear to be less studied, both in a natural product capacity, and as a source of crude extract bioactivity, yielding 9 and 5 species that display crude extract or pure compound bioactivity, respectively. Many studies have contributed to the isolation and identification of the large number of compounds that have been chemically profiled and discovered utilising hyphenated techniques such as HPLC–NMR and HPLC–MS [57,62,91,118,122,133,157,189]. This has expedited the process of dereplication and allowed for a more efficient pathway to the isolation and characterization of bioactive components present in crude extracts. As there can often be issues with the instability of compounds isolated from marine sources, these techniques limit the amount of exposure to the atmosphere or light that any purified sample would have by, promptly analyzing these samples after separation. Further use of such techniques could see an increase in the total number of natural products discovered, particularly from marine sources. Many studies included in this review appear to approach the isolation and characterization of natural products from a chemical perspective rather than by means of a bioassay guided fractionation. This provides an opportunity for utilizing these methodologies to attempt a more targeted natural product isolation, with the aim of furnishing more information about the bioactivity of crude extracts from marine organisms, and so increase the number of species studied.

## Figures and Tables

**Figure 1 marinedrugs-18-00142-f001:**
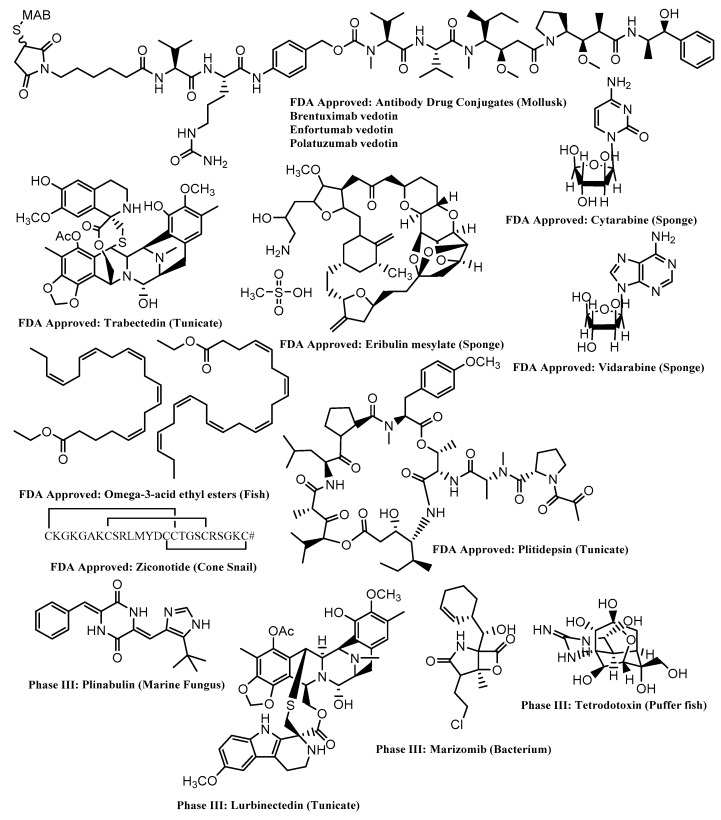
Food and Drug Administration (FDA)-approved and Phase III clinical drug candidates derived from marine natural products.

**Figure 2 marinedrugs-18-00142-f002:**
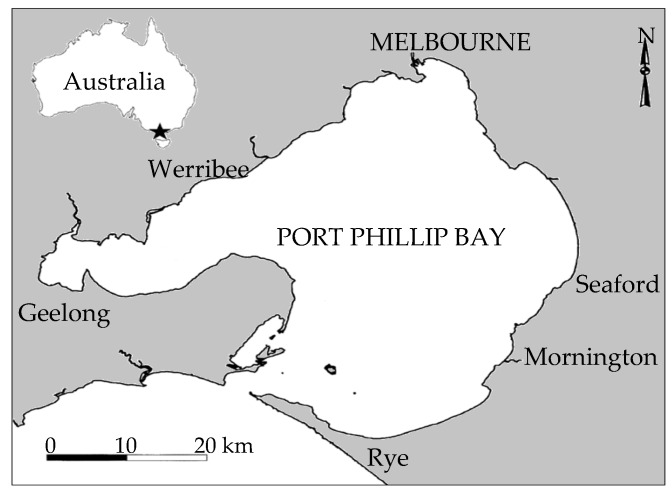
Geographical location of Port Phillip Bay in relation to Australia.

**Figure 3 marinedrugs-18-00142-f003:**
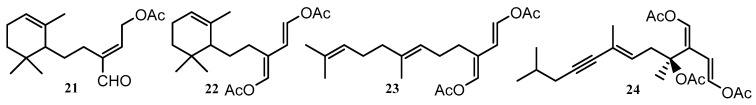
Chemical structure of sesquiterpenes **21**–**24**.

**Figure 4 marinedrugs-18-00142-f004:**
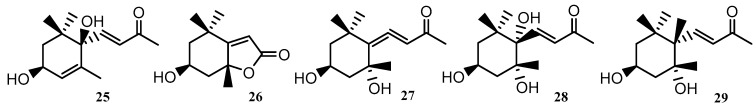
Chemical structure of cyclic geranyl acetones **25**–**29**.

**Figure 5 marinedrugs-18-00142-f005:**
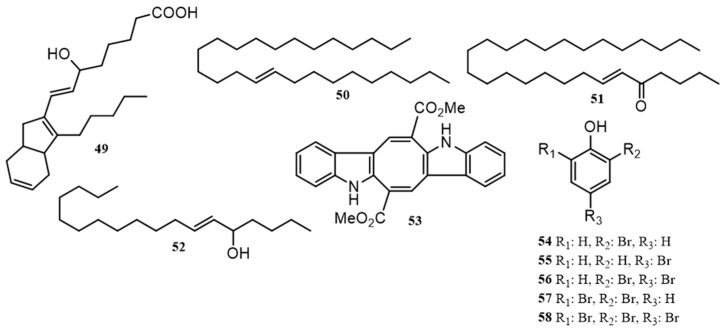
Chemical structure of miscellaneous compounds **49**–**58**.

**Figure 6 marinedrugs-18-00142-f006:**
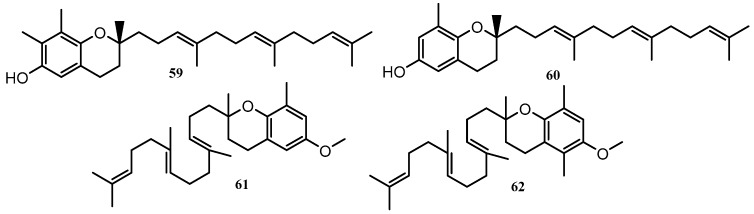
Chemical structure of tocotrienols **59**–**62**.

**Figure 7 marinedrugs-18-00142-f007:**
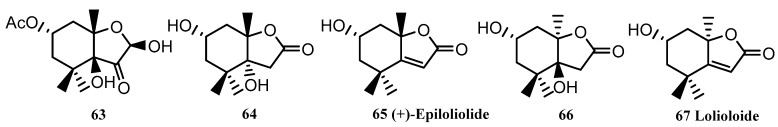
Chemical structure of monoterpenes **63**–**67**.

**Figure 8 marinedrugs-18-00142-f008:**
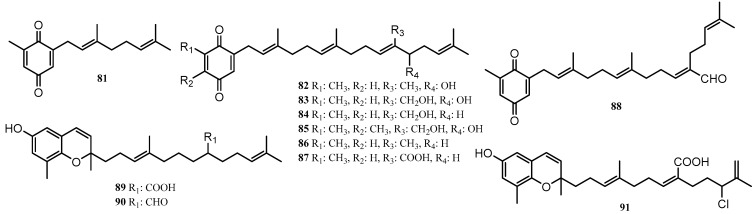
Chemical structure of Meroditerpenoids **81**–**91**.

**Figure 9 marinedrugs-18-00142-f009:**
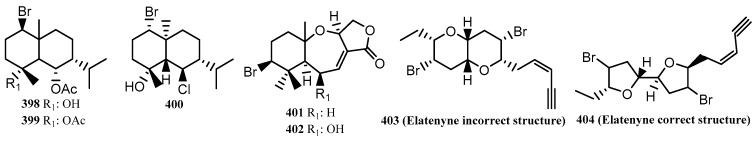
Chemical structure of sesquiterpenes **398**–**404**.

**Figure 10 marinedrugs-18-00142-f010:**
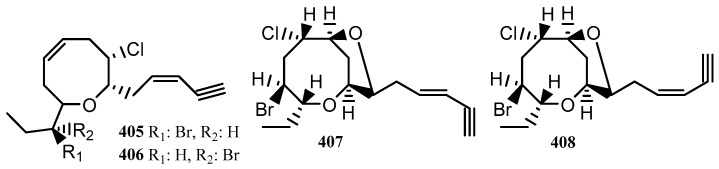
Chemical structure of lauroxocanes **405**–**408**.

**Figure 11 marinedrugs-18-00142-f011:**
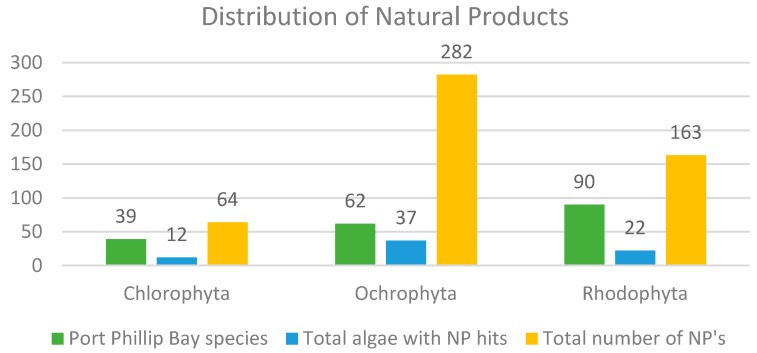
Total number of natural products isolated from marine algae common to Port Phillip Bay, represented by phylum.

**Figure 12 marinedrugs-18-00142-f012:**
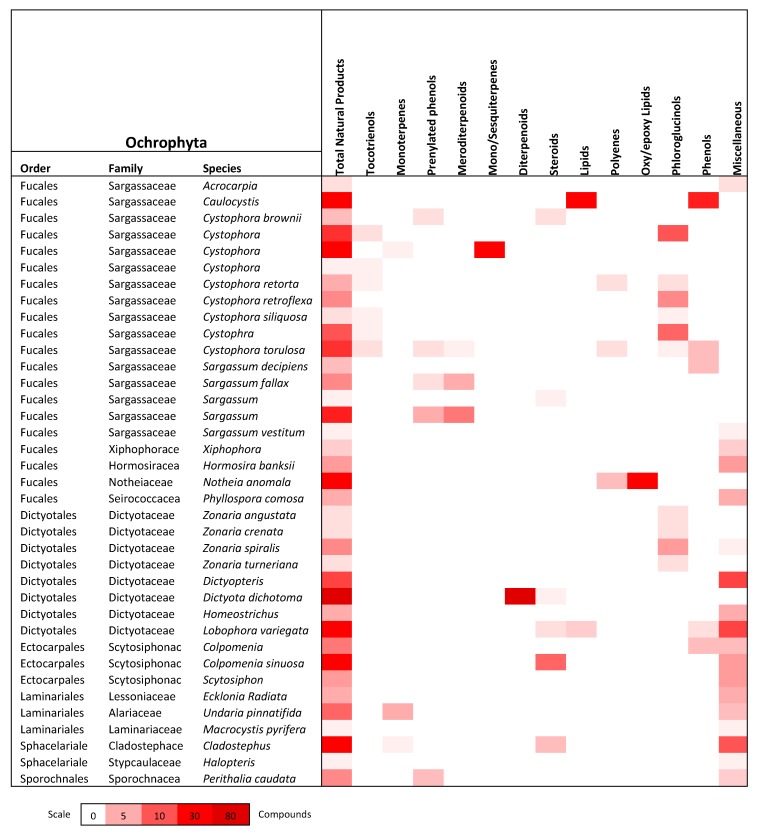
Heat maps of compound class distribution for Ochrophyta algae.

**Figure 13 marinedrugs-18-00142-f013:**
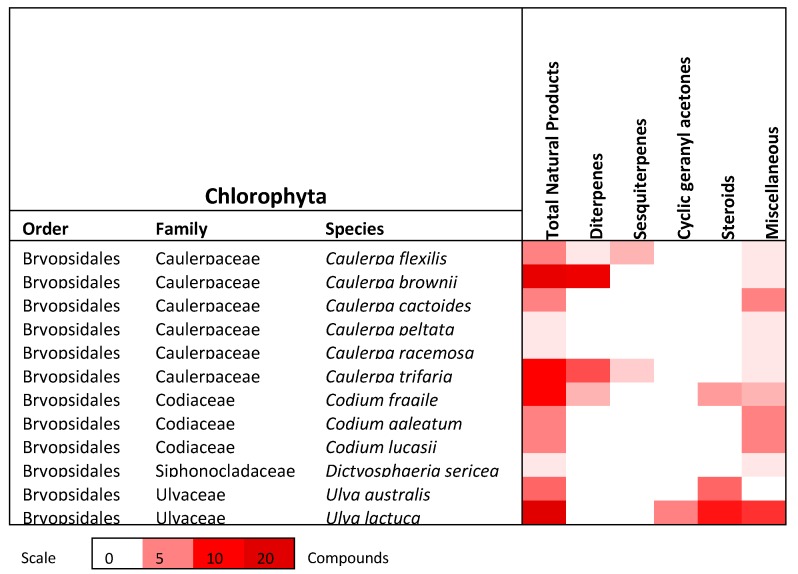
Heat maps of compound class distribution for Chlorophyta algae.

**Figure 14 marinedrugs-18-00142-f014:**
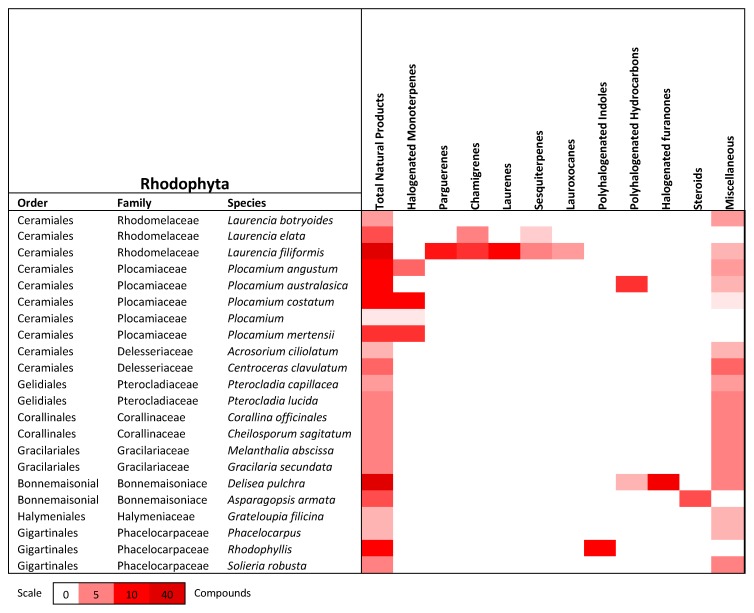
Heat maps of compound class distribution for Rhodophyta algae.

**Table 1 marinedrugs-18-00142-t001:** Reported marine algae species of the Port Phillip Bay region [21,22].

**CHLOROPHYTA**	*■Carpoglossum confluens*	*■Sargassum fallax*	*■Dasyphloea insignis*	*■Plocamium angustum*
*▲Apjohnia laetevirens*	*■Chlanidophora microphylla*	*■Sargassopsis heteromorphum*	*▲Delisea hypneoides*	*■Plocamium cirrhosum*
*▲Bryopsis vestita*	*■Cladostephus spongiosus*	*■Sargassum linearifolium*	*■Dictyomenia harveyana*	*▲Plocamium dilatatum*
*■Caulerpa alternans*	*■Colpomenia sinuosa*	*■Sargassum paradoxum*	*■Dictyomenia tridens*	*■Plocamium mertensii*
*■Caulerpa brownie*	*■Cystophora brownii*	*■Sargassum sonderi*	*■Diplocladia patersonis*	*▲Plocamium patagiatum*
*■Caulerpa cactoides*	*■Cystophora expansa*	*■Sargassum spinuligerum*	*▲Dudresnaya australis*	*■Plocamium preissianum*
*▲Caulerpa cliftonii*	*■Cystophora grevillei*	*■Sargassum vestitum*	*■Echinothamnion hystrix*	*■Pollexfenia lobata*
*■Caulerpa flexilis*	*■Cystophora moniliformis*	*■Scytosiphon lomentaria*	*■Echinothamnion mallardiae*	*■Pollexfenia pedicellata*
*■Caulerpa geminata*	*■Cystophora monilifera*	*■Seirococcus axillaris*	*▲Erythroclonium sonderi*	*■Polysiphonia decipiens*
*▲Caulerpa longifolia*	*■Cystophora platylobium*	*■Sirophysalis trinodis*	*▲Erythroclonium muelleri*	*■Polyopes constrictus*
*■Caulerpa obscura*	*■Cystophora retorta*	*■Suringariella harveyana*	*▲Gelidium australe*	*■Pterocladia lucida*
*▲Caulerpa papillosa*	*■Cystophora retroflexa*	*■Undaria pinnatifida*	*■Gelidium asperum*	*■Rhodoglossum gigartinoides*
*▲Caulerpa remotifolia*	*▲Cystophora siliquosa*	*■Xiphophora chondrophylla*	*■Gelinaria ulvoidea*	*■Rhodymenia leptophylla*
*■Caulerpa scalpelliformis*	*■Cystophora subfarcinata*	*■Zonaria angustata*	*▲Gigartina sonderi*	*▲Rhodymenia novaehollandica*
*■Caulerpa sedoides*	*■Cystophora torulosa*	*■Zonaria crenata*	*■Gracilaria cliftonii*	*■Rhodymenia obtusa*
*■Caulerpa simpliciuscula*	*■Caulocystis uvifera*	*■Zonaria spiralis*	*♦Grateloupia filicina*	*▲Rhodymenia prolificans*
*▲Caulerpa trifaria*	*■Dictyota dichotoma*	*▲Zonaria turneriana*	*■Halopeltis australis*	*■Sarcothalia crassifolia*
*■Caulerpa vesiculifera*	*■Dictyota furcellata*	**RHODOPHYTA**	*▲Halymenia plana*	*■Sonderophycus capensis*
*■Chaetomorpha coliformis*	*■Dictyota gunniana*	*■Acrosorium ciliolatum*	*■Hemineura frondosa*	*■Sonderophycus coriaceus*
*■Chaetomorpha linum*	*▲Dictyota paniculata*	*■Acrotylus australis*	*■Heterosiphonia gunniana*	*■Stenogramma interruptum*
*■Chaetomorpha valida*	*■Dictyopteris acrostichoides*	*■Agarophyton chilense*	*■Heterosiphonia muelleri*	*■Thuretia quercifolia*
*■Chlorodesmis baculifera*	*■Dictyopteris muelleri*	*■Ahnfeltiopsis fastigiata*	*■Hypnea ramentacea*	*■Wrangelia nobilis*
*■Cladophora prolifera*	*▲Distromium flabellatum*	*■Areschougia congesta*	*■Jania rosea*	*■Wrangelia plumosa*
*▲Cladophora rhizoclonioidea*	*■Durvillaea potatorum*	*■Ballia callitricha*	*■Jania sagittata*	
*▲Codium australicum*	*■Ecklonia radiata*	*▲Botryocladia sonderi*	*■Laurencia botryoides*	
*▲Codium duthieae*	*■Exallosorus olsenii*	*■Callophycus laxus*	*♦Laurencia elata*	
*■Codium fragile*	*■Halopteris pseudospicata*	*■Callophyllis rangiferina*	*■Laurencia filiformis*	
*■Codium galeatum*	*■Homoeostrichus sinclairii*	*■Callophyllis lambertii*	*▲Lenormandia marginata*	
*▲Codium harveyi*	*■Hormosira banksii*	*■Camontagnea oxyclada*	*■Lenormandia muelleri*	
*■Codium lucasii*	*■Leathesia marina*	*▲Capreolia implexa*	*■Lophurella periclados*	
*▲Codium pomoides*	*■Lobospira bicuspidata*	*■Centroceras clavulatum*	*■Martensia australis*	
*▲Dictyosphaeria sericea*	*■Lobophora variegata*	*■Cephalocystis furcellata*	*■Melanthalia obtusata*	
*■Ulva australis*	*■Macrocystis pyrifera*	*■Champia viridis*	*■Meredithia nana*	
*■Ulva compressa*	*▲Notheia anomala*	*■Chrysymenia brownii*	*■Metagoniolithon radiatum*	
*■Ulva lactuca*	*■Padina fraseri*	*■Corallina officinale*	*■Metamastophora flabellata*	
*■Ulva linza*	*■Perithalia caudata*	*■Corynecladia clavata*	*■Nizymenia australis*	
*■Ulva rigida*	*■Petalonia fascia*	*■Crassiphycus secundatus*	*■Pachymenia orbicularis*	
*■Ulva taeniata*	*■Phyllospora comosa*	*▲Cryptonemia undulata*	*■Palisada tumida*	
**OCHROPHYTA**	*■Phyllotricha decipiens*	*■Curdiea angustata*	*■Perbella minuta*	
*■Acrocarpia paniculata*	*■Phyllotricha varians*	*■Dasya ceramioides*	*▲Phacelocarpus alatus*	
*▲Bellotia eriophorum*	*■Phyllotricha verruculosa*	*■Dasya naccarioides*	*▲Phacelocarpus complanatus*	
*■Caulocystis cephalornithos*	*♦Sargassum decipiens*	*■Dasya wilsonis*	*■Phacelocarpus peperocarpos*	

♦ Currently unaccepted name but is mentioned in the literature frequently; see Appendix A for new name. *■* Species that are the currently accepted name and have other names that are synonyms. ▲ Species that are the currently accepted name and have no synonyms.

**Table 2 marinedrugs-18-00142-t002:** Distribution of compounds **1** to **29**.

No.	Compound Type	Species	Origin	Ref
**1, 2**	Diterpene	*C. brownii*	Spring Beach, East Tasmania	[27]
**2a–f**	Diterpene	*C. brownii*	Spring Beach, East Tasmania	[27]
**3, 4**	Diterpene	*C. trifaria*	Taroona Beach, Hobart, Tasmania	[25]
**5**	Diterpene	*C. trifaria*	Taroona Beach, Hobart, Tasmania	[25,27]
		*C. brownii*	Spring Beach, East Tasmania	
**6**	Diterpene	*C. brownii*	Spring Beach, East Tasmania	[27]
**7**	Diterpene	*C. brownii*	Flinders Reef, Victoria	[27]
**8**	Diterpene	*C. brownii*	Spring Beach, East Tasmania	[27]
**9**	Diterpene	*C. trifaria*	Taroona Beach, Hobart, Tasmania	[25]
**10**	Diterpene	*C. brownii*	Spring Beach, East Tasmania	[27]
**11**	Diterpene	*C. trifaria*	Taroona Beach, Hobart, Tasmania	[25]
**12**	Diterpene	*C. brownii*	Spring Beach, East Tasmania	[27]
**13–15**	Diterpene	*C. fragile*	Qingdao Coastline, Shangdong, China	[28]
**16**	Diterpene	*C. brownii*	Spring Beach, East Tasmania	[27]
**17**	Diterpene	*C. trifaria*	Taroona Beach, Hobart, Tasmania	[25,36]
		*C. flexilis*	Cosy corner, Western Australia	
**18**	Diterpene	*C. trifaria*	Taroona Beach, Hobart, Tasmania	[25,27]
		*C. brownii*	Spring Beach, East Tasmania	
**19, 20**	Diterpene	*C. brownii*	Spring Beach, East Tasmania	[27]
**21, 22**	Sesquiterpene	*C. flexilis*	Cosy corner, Western Australia	[36]
**23**	Sesquiterpene	*C. flexilis*	Cosy corner, Western Australia	[36,37]
		*C. trifaria*	-	
**24**	Sesquiterpene	*C. trifaria*	Taroona Beach, Hobart, Tasmania	[25]
**25–29**	Cyclic geranylacetone	*U. lactuca*	BoHai Coastline, China	[38]

**Table 3 marinedrugs-18-00142-t003:** Distribution of compounds **30** to **58**.

No.	Compound Type	Species	Origin	Ref
**30–36**	Steroid/Sterol	*U. lactuca*	Bay of Kotor, Southern Adriatic Sea	[40]
**37**	Steroid/Sterol	*C. fragile*	-	[45]
**38, 39**	Steroid/Sterol	*C. fragile*	Qingdao Coastline, Shangdong, China	[28]
**40**	Steroid/Sterol	*U. lactuca*	Abu Qir Bay, Alexandria, Egypt	[40,52]
			Bay of Kotor, Southern Adriatic Sea	
**41–46**	Steroid/Sterol	*U. australis*	Dalian Coast, China	[39]
**47**	Steroid/Sterol	*U. lactuca*	Abu Qir Bay, Alexandria, Egypt	[41]
**48**	Steroid/Sterol	*C. fragile*	Qingdao Coastline, Shangdong, China	[28]
**49**	Lipid	*D. sericea*	Cape Schank and Point Lonsdale, Victoria	[51]
**50–52**	Lipid	*U. lactuca*	Abu Qir Bay, Alexandria, Egypt	[41]
**53**	Di indolo pigment	*C. trifaria*	Point Peron, WA	[50]
		*C. brownii*	Augusta, WA	
		*C. flexilis*	Augusta, WA	
		*C. peltata*	Big Nook Island, WA	
		*C. racemosa*	Big Nook Island, WA	
**54**	Bromophenolic	*U. lactuca*	Bateau Bay, NSW	[54]
		*C. lucasii*	Bateau Bay, NSW	
		*C. galeatum*	Bateau Bay, NSW	
		*C. cactoides*	Bateau Bay, NSW	
**55**	Bromophenolic	*U. lactuca*	Bateau Bay, NSW	[54]
		*C. lucasii*	Bateau Bay, NSW	
		*C. galeatum*	Bateau Bay, NSW	
		*C. cactoides*	Bateau Bay, NSW	
**56**	Bromophenolic	*U. lactuca*	Bateau Bay, NSW	[54]
		*C. lucasii*	Bateau Bay, NSW	
		*C. galeatum*	Bateau Bay, NSW	
		*C fragile*	Bateau Bay, NSW	
		*C. cactoides*	Bateau Bay, NSW	
**57**	Bromophenolic	*U. lactuca*	Bateau Bay, NSW	[54]
		*C. lucasii*	Bateau Bay, NSW	
		*C. galeatum*	Bateau Bay, NSW	
		*C. fragile*	Bateau Bay, NSW	
		*C. cactoides*	Bateau Bay, NSW	
**58**	Bromophenolic	*U. lactuca*	Bateau Bay, NSW	[54]
		*C. lucasii*	Bateau Bay, NSW	
		*C. galeatum*	Bateau Bay, NSW	
		*C fragile*	Bateau Bay, NSW	
		*C. cactoides*	Bateau Bay, NSW	

**Table 4 marinedrugs-18-00142-t004:** Distribution of compounds **59** to **80**.

No.	Compound Type	Species	Origin	Ref
**59**	Tocotrienols	*C. monilifera*	Governor Reef, Indented Head, Victoria	[57]
**60**	Tocotrienols	*C. subfarcinata*	Queenscliffe, Victoria	[55,56,57,58]
		*C. platylobium*	-	
		*C. monilifera*	Governor Reef, Indented Head, Victoria	
		*C. siliquosa*	Sorrento Back Beach, Victoria	
		*C. retorta*	Cowaramup Bay, WA	
**61, 62**	Tocotrienols	*C. torulosa*	Torquay, Victoria	[55]
**63–66**	Monoterpenes	*U. pinnatifida*	Miura Peninsula, Japan	[59]
**67**	Monoterpenes	*U. pinnatifida*	Miura Peninsula, Japan	[59,68,69]
		*C. moniliformis*	-	
		*C. spongiosus*	Tipaza, Algerian Mediterranean Coast	
**68, 69**	Prenylated Phenols	*S. paradoxum*	Governor Reef, Indented Head, Victoria	[62,63]
		*S. fallax*	Governor Reef, Indented Head, Victoria	
**70, 71**	Prenylated Phenols	*S. paradoxum*	Governor Reef, Indented Head, Victoria	[57]
**72**	Prenylated Phenols	*P. caudata*	Flinders Reef, Victoria	[66]
**73**	Prenylated Phenols	*P. caudata*	Ninepin point, D’Entrecasteaux Channel, Tasmania	[67]
**74**	Prenylated Phenols	*P. caudata*	Flinders Reef, Victoria	[66]
**75, 76**	Prenylated Phenols	*C. brownii*	Victor Harbour, SA	[56]
**77**	Prenylated Phenols	*S. paradoxum*	Governor Reef, Indented Head, Victoria	[57]
**78, 79**	Prenylated Phenols	*C. torulosa*	Cook Straight, Wellington, New Zealand	[58]
**80**	Prenylated Phenols	*P. caudata*	Flinders Reef, Victoria	[66]

**Table 5 marinedrugs-18-00142-t005:** Distribution of compounds **81** to **111**.

No.	Compound Type	Species	Origin	Ref
**81**	Meroditerpenoids	*C. torulosa*	Cook Straight, Wellington, New Zealand	[58]
**82**	Meroditerpenoids	*S. paradoxum*	Governor Reef, Indented Head, Victoria	[57]
**83**	Meroditerpenoids	*S. paradoxum*	Governor Reef, Indented Head, Victoria	[57,63]
		*S. fallax*	Governor Reef, Indented Head, Victoria	
**84, 85**	Meroditerpenoids	*S. paradoxum*	Governor Reef, Indented Head, Victoria	[57]
**86, 87**	Meroditerpenoids	*S. paradoxum*	Governor Reef, Indented Head, Victoria	[57,63]
		*S. fallax*	Governor Reef, Indented Head, Victoria	
**88**	Meroditerpenoids	*S. paradoxum*	Governor Reef, Indented Head, Victoria	[57]
**89**	Meroditerpenoids	*S. fallax*	Governor Reef, Indented Head, Victoria	[63]
**90**	Meroditerpenoids	*S. paradoxum*	Governor Reef, Indented Head, Victoria	[57]
**91**	Meroditerpenoids	*S. fallax*	Governor Reef, Indented Head, Victoria	[63]
**92–94**	Farnesylacetone epoxide	*C. moniliformis*	Sarge Bay, Cape Leeuwin, WA	[58]
**95–97**	Farnesylacetone	*C. moniliformis*	Port Phillip Bay, Victoria	[72]
**98–100**	Geranylacetone, Geranylgeranal epoxide	*C. moniliformis*	North East side of West Island, SA	[70]
**101**	Farnesylacetone	*C. moniliformis*	-	[71]
**102, 103**	Cyclic farnesylacetone	*C. moniliformis*	Port Phillip Bay, Victoria	[72]
**104, 105**	Cyclic farnesylacetone	*C. moniliformis*	-	[68]
**106, 107**	Cyclic farnesylacetone	*C. moniliformis*	Port Phillip Bay, Victoria	[72]
**108**	Cyclic farnesylacetone	*C. moniliformis*	-	[68]
**109**	Cyclic farnesylacetone	*C. moniliformis*	North East side of West Island, SA	[70]
**110**	Cyclic farnesylacetone	*C. moniliformis*	Port Phillip Bay, Victoria	[72]
**111**	Aromadendrene	*C. moniliformis*	-	[68]

**Table 6 marinedrugs-18-00142-t006:** Distribution of compounds **112** to **192**.

No.	Compound Type	Species	Origin	Ref
**112**	Diterpene	*D. dichotoma*	Northern Adriatic Sea	[73]
**113**	Diterpene	*D. dichotoma*	Red Sea	[73]
**114**	Diterpene	*D. dichotoma*	Saronicos Gulf, Greece	[73]
**115–117**	Diterpene	*D. dichotoma*	Tyrrhenian Sea	[73]
**118**	Diterpene	*D. dichotoma*	Puerto Madryn	[73]
**119**	Diterpene	*D. dichotoma*	Red Sea	[73]
**120**	Diterpene	*D. dichotoma*	Northern Adriatic Sea	[73]
**121**	Diterpene	*D. dichotoma*	Tyrrhenian Sea	[73]
**122–124**	Diterpene	*D. dichotoma*	Red Sea, Egypt	[73]
**125**	Diterpene	*D. dichotoma*	Troitsa Bay, Russian Far East	[73]
**126–128**	Diterpene	*D. dichotoma*	Red Sea, Egypt	[73]
**129**	Diterpene	*D. dichotoma*	Japan	[73]
**130**	Diterpene	*D. dichotoma*	Red Sea, Egypt	[73]
**131, 132**	Diterpene	*D. dichotoma*	Patagonia	[73]
**133**	Diterpene	*D. dichotoma*	Tyrrhenian Sea	[73]
**134**	Diterpene	*D. dichotoma*	-	[73]
**135**	Diterpene	*D. dichotoma*	Acicastello, Italy	[73]
**136, 137**	Diterpene	*D. dichotoma*	Russian Far East	[73]
**138, 140**	Diterpene	*D. dichotoma*	-	[73]
**141, 142**	Diterpene	*D. dichotoma*	Acicastello, Italy	[73]
**143**	Diterpene	*D. furcellata*	Cape Peron, Shark Bay, WA	[73]
**144–151**	Diterpene	*D. dichotoma*	Indian Ocean	[73]
**152–157**	Diterpene	*D. dichotoma*	Acicastello, Italy	[73]
**158–164**	Diterpene	*D. dichotoma*	Indian Ocean	[73]
**165, 166**	Diterpene	*D. dichotoma*	Red Sea	[73]
**167, 168**	Diterpene	*D. dichotoma*	-	[73]
**169, 171**	Diterpene	*D. dichotoma*	Karachi Coast, Arabian Sea	[73]
**172, 173**	Diterpene	*D. dichotoma*	Red Sea	[73]
**174**	Diterpene	*D. dichotoma*	Indian Ocean	[73]
**175**	Diterpene	*D. dichotoma*	-	[73]
**176**	Diterpene	*D. dichotoma*	Oshoro Bay, Hokkaido, Japan	[73]
**177–179**	Diterpene	*D. dichotoma*	Yagachi, Okinawa, Japan	[73]
**180**	Diterpene	*D. dichotoma*	-	[73]
**181–183**	Diterpene	*D. dichotoma*	Oshoro Bay, Hokkaido, Japan	[73]
**184–186**	Diterpene	*D. dichotoma*	-	[73]
**187**	Diterpene	*D. dichotoma*	Yagachi, Okinawa, Japan	[73]
**188**	Diterpene	*D. dichotoma*	-	[73]
**189**	Diterpene	*D. dichotoma*	Nagahama Beach, Ehime, Japan	[73]
**190, 191**	Diterpene	*D. dichotoma*	Troitsa Bay, Russian Far East	[73]
**192**	Diterpene	*D. dichotoma*	-	[73]

**Table 7 marinedrugs-18-00142-t007:** Distribution of compounds **193** to **254**.

No.	Compound Type	Species	Origin	Ref
**193–195**	Steroids/Sterols	*C. sinuosa*	Cap Vert, Dakar	[76]
		*D. dichotoma*	-	
**196**	Steroids/Sterols	*C. sinuosa*	Cap Vert, Dakar	[43,76]
		*C. spongiosus*	Praia do quebrado, Portugal	
		*D. dichotoma*	-	
**197**	Steroids/Sterols	*C. sinuosa*	Cap Vert, Dakar	[76]
		*D, dichotoma*		
**198, 199**	Steroids/Sterols	*C. sinuosa*	Cap Vert, Dakar	[42,43,76]
		*C. spongiosus*	Praia do quebrado, Portugal	
		*L. variegata*	St Thomas, Virgin Islands	
		*D. dichotoma*	Cap Vert, Dakar	
**200**	Steroids/Sterols	*S. linearfolium*	Bateau Bay, NSW	[43,75,76]
		*C. sinuosa*	Cap Vert, Dakar	
		*D. dichotoma*	Cap Vert, Dakar	
		*C. spongiosus*	Praia do quebrado, Portugal	
**201**	Steroids/Sterols	*C. brownii*	Victor Harbour, SA	[56]
**202**	Steroids/Sterols	*C. brownii*	Victor Harbour, SA	[56,76]
		*C. sinuosa*	Cap Vert, Dakar	
**203–221**	Lipid	*C. cephalornithos*	Southern and South Eastern Tasmania	[79]
**222**	Lipid	*C. cephalornithos*	Victorian Coastline	[87]
**223**	Lipid	*C. cephalornithos*	Southern and South Eastern Tasmania	[79]
**224**	Lipid	*C. cephalornithos*	Victorian Coastline	[87]
**225–227**	Lipid	*L. variegata*	Tenerife, Canary Islands	[80]
**228**	Polyene	*N. anomala*	Bells Beach, Victoria	[55,58,86]
		*C. torulosa*	Torquay, Victoria	
		*C. retorta*	Cowaramup Bay, WA	
**229, 230**	Polyene	*N. anomala*	Bells Beach, Victoria	[86]
**231**	Polyene	*N. anomala*	Bells Beach, Victoria	[58,86]
		*C. torulosa*	Cook Straight, Wellington, New Zealand	
		*C. retorta*	Cowaramup Bay, WA	
**232, 233**	Oxylipid	*N. anomala*	Bells Beach, Victoria	[84]
**234**	Epoxylipid	*N. anomala*	Torquay, Victoria	[82]
**235**	Epoxylipid	*N. anomala*	Southern and South Eastern Tasmania	[88]
**236–245**	Epoxylipid	*N. anomala*	Bells Beach, Victoria	[86,89]
**246–254**	Oxylipid	*N. anomala*	Bells Beach, Victoria	[86,89]

**Table 8 marinedrugs-18-00142-t008:** Distribution of compounds **255** to **297**.

No.	Compound Type	Species	Origin	Ref
**255**	Phloroglucinol	*C. subfarcinata*	Queenscliffe, Victoria	[57,58,91]
		*C. monilifera*	Governor Reef, Indented Head, Victoria	
		*C. retroflexa*	Governor Reef, Indented Head, Victoria	
		*C. retorta*	Cowaramup Bay, WA	
**256**	Phloroglucinol	*C. torulosa*	Torquay, Victoria	[55,57,58]
		*C. subfarcinata*	Queenscliffe, Victoria	
		*C. siliquosa*	Sorrento Back Beach	
		*C. retorta*	Cowaramup Bay, WA	
		*C. monilifera*	Governor Reef, Indented Head, Victoria	
**257, 258**	Phloroglucinol	*Z. spiralis*	North Walkerville, Victoria	[93]
**259**	Phloroglucinol	*C. subfarcinata*	Queenscliffe, Victoria	[57]
		*C. monilifera*	Governor Reef, Indented Head, Victoria	
**260**	Phloroglucinol	*C. subfarcinata*	Queenscliffe, Victoria	[57]
**261**	Phloroglucinol	*C. subfarcinata*	North Eastern West Island, SA	[58,92]
		*C. monilifera*	-	
**262**	Phloroglucinol	*C. retroflexa*	Governor Reef, Indented Head, Victoria	[57,91]
		*C. monilifera*	Governor Reef, Indented Head, Victoria	
**263**	Phloroglucinol	*C. retroflexa*	Governor Reef, Indented Head, Victoria	[91]
**264**	Phloroglucinol	*Z. spiralis*	North Walkerville, Victoria	[93]
**265**	Phloroglucinol	*C. subfarcinata*	Queenscliffe, Victoria	[57,91]
		*C. monilifera*	Governor Reef, Indented Head, Victoria	
		*C. retroflexa*	Governor Reef, Indented Head, Victoria	
**266**	Phloroglucinol	*C. monilifera*	Governor Reef, Indented Head, Victoria	[57]
**267, 268**	Phloroglucinol	*Z. turneriana*	Tinderbox, Tasmania	[94]
		*Z. crenata*	Tinderbox, Tasmania	
		*Z angustata*	Sisters Beach, Tasmania	
**269, 270**	Phloroglucinol	*C. subfarcinata*	Queenscliffe, Victoria	[57,91]
		*C. monilifera*	Governor Reef, Indented Head, Victoria	
		*C. retroflexa*	Governor Reef, Indented Head, Victoria	
**271**	Phloroglucinol	*C. subfarcinata*	Queenscliffe, Victoria	[57,91]
		*C. retroflexa*	Governor Reef, Indented Head, Victoria	
**272**	Phloroglucinol	*C. monilifera*	Governor Reef, Indented Head, Victoria	[57]
**273–275**	Phloroglucinol	*Z. spiralis*	North Walkerville, Victoria	[93]
**276, 277**	Benzopyranones	*C. cephalornithos*	Southern and South Eastern Tasmania	[79]
**278**	Phenolic Acid	*C. cephalornithos*	Southern and South Eastern Tasmania	[79,91]
		*S. decipiens*	Governor Reef, Indented Head, Victoria	
**279**	Phenolic Acid	*C. cephalornithos*	Southern and South Eastern Tasmania	[79]
**280**	Phenolic Acid	*C. cephalornithos*	Southern and South Eastern Tasmania	[79,91]
		*S. decipiens*	Governor Reef, Indented Head, Victoria	
**281, 282**	Phenolic Acid	*C. cephalornithos*	Southern and South Eastern Tasmania	[79]
**283**	Phenol	*C. cephalornithos*	Southern and South Eastern Tasmania	[79,91]
		*S. decipiens*	Governor Reef, Indented Head, Victoria	
**284**	Phenol	*C. cephalornithos*	Southern and South Eastern Tasmania	[79]
**285**	Resorcinol	*C. cephalornithos*	Southern and South Eastern Tasmania	[79,91]
		*S. decipiens*	Governor Reef, Indented Head, Victoria	
**286**	Resorcinol	*C. cephalornithos*	Southern and South Eastern Tasmania	[55,79]
		*C. torulosa*	Torquay, Victoria	
**287**	Resorcinol	*C. torulosa*	Torquay, Victoria	[55]
**288, 289**	Phenolic Acid	*C. cephalornithos*	Southern and South Eastern Tasmania	[79]
**290**	Resorcinol	*C. torulosa*	Cook Straight, Wellington, New Zealand	[58]
**291**	Resorcinol	*C. torulosa*	Torquay, Victoria	[55]
**292–295**	Phenolic Acid	*C. peregrina*	Bulgarian Coast	[100]
**296, 297**	Phenolic Acid	*L. variegata*	Tenerife, Canary Islands	[80]

**Table 9 marinedrugs-18-00142-t009:** Distribution of compounds **298** to **339**.

No.	Compound Type	Species	Origin	Ref
**298**	Xestoaminol	*X. chondrophylla*	Hen and Chicken Islands, New Zealand	[101]
**299–303**	Pheromone	*D. acrostichoides*	Point Lonsdale and Sorrento, Victoria	[102]
**304, 305**	Pheromone	*D. acrostichoides*	Point Lonsdale and Sorrento, Victoria	[102]
		*C. peregrina*	Flinders, Victoria	[106]
**306**	Pheromone	*D. acrostichoides*	Point Lonsdale and Sorrento, Victoria	[102]
**307**	Pheromone	*X. chondrophylla*	-	[105]
		*S. lomentaria*	-	
**308**	Pheromone	*H. banksii*	Flinders Reef, Victoria	[103]
**309**	Pheromone	*X. chondrophylla*	-	[105]
		*S. lomentaria*	-	
**310–312**	Pheromone	*P. caudata*	-	[114]
**313**	Pheromone	*C. spongiosus*	Flinders Reef, Victoria	[102,115]
		*D. acrostichoides*	Point Lonsdale and Sorrento, Victoria	
**314**	Pheromone	*M. pyrifera*	-	[103]
		*U. pinnatifida*	-	
**315**	Pheromone	*C. spongiosus*	Flinders Reef, Victoria	[115]
**316, 317**	Pheromone	*D. acrostichoides*	Point Lonsdale and Sorrento, Victoria	[102]
**318–322**	Bromophenolic	*C. spongiosus*	Bateau Bay, NSW	[54]
		*C. sinuosa*	Bateau Bay, NSW	
		*E. radiata*	Bateau Bay, NSW	
		*H. sinclairii*	Bateau Bay, NSW	
		*H. banksii*	Bateau Bay, NSW	
		*P. comosa*	Bateau Bay, NSW	
		*L. variegata*	Bateau Bay, NSW	
**323**	Bromophenolic	*C. sinuosa*	Gulf of Eilat, Israel	[107]
**324**	Xanthophyll	*S. lomentaria*	Aikappu, Akkeshi, Hokkaido	[111]
**325**	Xanthophyll	*S. lomentaria*	Aikappu, Akkeshi, Hokkaido	[69,111]
		*C. spongiosus*	Algerian Mediterranean Coast, Tipaza	
**326**	Xanthophyll	*S. lomentaria*	Aikappu, Akkeshi, Hokkaido	[69,93,111]
		*C. spongiosus*	Algerian Mediterranean Coast, Tipaza	
		*Z. spiralis*	North Walkerville, Victoria	
**327, 328**	Xanthophyll	*U. pinnatifida*	-	[108]
**329**	Xanthophyll	*U. pinnatifida*	-	[69,91,108,111]
		*S. lomentaria*	Aikappu, Akkeshi, Hokkaido	
		*C. spongiosus*	Algerian Mediterranean Coast, Tipaza	
		*H. pseudospicata*	Queenscliffe, Victoria	
		*S. vestitum*	Queenscliffe, Victoria	
**330, 331**	Furans	*A. paniculata*	Port MacDonnell	[116]
**332**	Pyridine	*C. peregrina*	Bulgarian Coast	[100]
**333**	Amine	*C. peregrina*	Bulgarian Coast	[100]
**334**	Polyketide Macrolide	*L. variegata*	Cay Lobos, Bahamas	[113]
**335–339**	Polyketides	*L. variegata*	Tenerife, Canary Islands	[80]

**Table 10 marinedrugs-18-00142-t010:** Distribution of compounds **340** to **385**.

No.	Compound Type	Species	Origin	Ref
**340–342**	Halogenated Monoterpene	*P. mertensii*	Queenscliffe, Victoria	[118]
**343**	Halogenated Monoterpene	*P. mertensii*	Carnac Island, WA	[139]
**344, 345**	Halogenated Monoterpene	*P. mertensii*	Queenscliffe, Victoria	[118]
**346**	Halogenated Monoterpene	*P. mertensii*	-	[117]
**347**	Halogenated Monoterpene	*P. mertensii*	Queenscliffe, Victoria	[118]
**348**	Halogenated Monoterpene	*P. angustum*	Cape Northumberland, SA	[140]
**349, 350**	Halogenated Monoterpene	*P. angustum*	Rocky Point, Torquay, Victoria	[141]
**351**	Halogenated Monoterpene	*P. angustum*	Queenscliffe, Victoria	[122,123]
		*P. costatum*	Robe, South Australia	
**352, 353**	Halogenated Monoterpene	*P. angustum*	Queenscliffe, Victoria	[122]
**354**	Halogenated Monoterpene	*P. costatum*	Robe, South Australia	[123]
**355**	Halogenated Monoterpene	*P. costatum*	Port MacDonnell, South Australia	[124]
**356**	Halogenated Monoterpene	*P. costatum*	Deep Glen Bay, Tasmania	[142]
**357**	Halogenated Monoterpene	*P. costatum*	Pandalowie Bay, South Australia	[125]
**358**	Halogenated Monoterpene	*P. costatum*	Pandalowie Bay, South Australia	[125]
**359, 360**	Halogenated Monoterpene	*P. costatum*	Deep Glen Bay, Tasmania	[142]
**361–364**	Halogenated Monoterpene	*P. costatum*	Pandalowie Bay, South Australia	[125]
**365**	Halogenated Monoterpene	*P. leptophyllum*	Toyama Bay, Japan	[126]
**366–374**	Parguerene	*L. filiformis*	South Australia	[127,128]
**375, 376**	Chamigrene	*L. elata*	St. Pauls Beach, Sorrento, Victoria	[133]
**377**	Chamigrene	*L. filiformis*	Taroona Beach, Hobart, Tasmania	[133,138]
		*L. elata*	St. Pauls Beach, Sorrento, Victoria	
**378**	Chamigrene	*L. elata*	New South Wales Coast	[132]
**379, 380**	Chamigrene	*L. filiformis*	Taroona Beach, Hobart, Tasmania	[138]
**381**	Chamigrene	*L. filiformis*	Taroona Beach, Hobart, Tasmania	[132,138]
		*L. elata*	New South Wales Coast	
**382–385**	Chamigrene	*L. filiformis*	Stella Maris Beach, Salvador, Brazil	[143]

**Table 11 marinedrugs-18-00142-t011:** Distribution of compounds **386** to **408**.

No.	Compound Type	Species	Origin	Ref
**386**	Laurene	*L. filiformis*	Hamelin Bay, Perth, WA	[164]
			Shoalwater Bay, Perth, WA	
			Cottesloe Beach, Perth, WA	
			Lancelin, Perth, WA	
**387, 388**	Laurene	*L. filiformis*	Shoalwater Bay, Perth, WA	[164]
			Cottesloe Beach, Perth, WA	
			Lancelin, Perth, WA	
**389**	Laurene	*L. filiformis*	South Australian Coast	[144]
**390**	Laurene	*L. filiformis*	St. Pauls Beach, Sorrento, Victoria	[146]
**391**	Laurene	*L. filiformis*	Shoalwater Bay, Perth, WA	[164]
**392**	Laurene	*L. filiformis*	Port MacDonnell Beach, South Australia	[145]
**393, 394**	Laurene	*L. filiformis*	Shoalwater Bay, Perth, WA	[164]
**395, 396**	Laurene	*L. filiformis*	St. Pauls Beach, Sorrento, Australia	[146]
**397**	Laurene	*L. filiformis*	Port MacDonnell Beach, South Australia	[145]
**398, 399**	Sesquiterpenoids	*L. filiformis*	Western Australia	[127,150]
			South Australia	
**400**	Sesquiterpenoids	*L. filiformis*	Port MacDonnell Beach, South Australia	[145,146]
			St. Pauls Beach, Sorrento, Australia	
**401, 402**	Sesquiterpenoids	*L. filiformis*	Point Peron, WA	[151]
**403, 404**	Sesquiterpenoids	*L. elata*	Batemans Bay, New South Wales	[133]
			St. Pauls Beach, Sorrento, Australia	
**405, 406**	Lauroxocane	*L. filiformis*	Western Australia	[150]
**407, 408**	Lauroxocane	*L. filiformis*	St. Pauls Beach, Sorrento, Australia	[133]

**Table 12 marinedrugs-18-00142-t012:** Distribution of compounds **409** to **473**.

No.	Compound Type	Species	Origin	Ref
**409–424**	Polyhalogenated Indole	*R. membranacea*	Moa Point, New Zealand	[165,166]
**425–426**	Polyhalogenated Hydrocarbon	*P. australasica*	Pearsons Point, Tasmania	[168]
**427**	Polyhalogenated Hydrocarbon	*D. pulchra*	Cape Banks, New South Wales	[167,169,170]
		*P. australasica*	Tasmania	
**428–430**	Polyhalogenated Hydrocarbon	*P. australasica*	Tasmania	[167]
**431**	Polyhalogenated Hydrocarbon	*D. pulchra*	Cape Banks, New South Wales	[169,170]
**432**	Polyhalogenated Hydrocarbon	*P. australasica*	Pearsons Point, Tasmania	[168]
**433**	Polyhalogenated Hydrocarbon	*P. australasica*	Tasmania	[167]
**434**	Polyhalogenated Hydrocarbon	*D. pulchra*	Cape Banks, New South Wales	[169]
**435**	Polyhalogenated Furanones	*D. pulchra*	Cape Banks, New South Wales	[169]
**436**	Polyhalogenated Furanones	*D. pulchra*	Cape Banks, New South Wales	[170,172]
			Palmer Station, Antarctica	
**437, 438**	Polyhalogenated Furanones	*D. pulchra*	Cape Banks, New South Wales	[169,170]
**439–442**	Polyhalogenated Furanones	*D. pulchra*	Cape Banks, New South Wales	[170,171]
**443**	Polyhalogenated Furanones	*D. pulchra*	Cape Banks, New South Wales	[170,171,172]
			Palmer Station, Antarctica	
**444–448**	Polyhalogenated Furanones	*D. pulchra*	Cape Banks, New South Wales	[170,171]
**449**	Polyhalogenated Furanones	*D. pulchra*	New South Wales	[174]
**450**	Polyhalogenated Furanones	*D. pulchra*	Cape Banks, New South Wales	[169]
**451–462**	Polyhalogenated Furanones	*D. pulchra*	Cape Banks, New South Wales	[171,175]
**463–465**	Polyhalogenated Furanones	*D. pulchra*	Palmer Station, Antarctica	[172]
**466**	Polyhalogenated Furanones	*D. pulchra*	Cape Banks, New South Wales	[169]
**467, 468**	Steroid	*A. armata*	-	[176]
**469–472**	Steroid	*A. armata*	Praia do Quebrado, Portugal	[43]
**473**	Steroid	*A. armata*	Portugal	[177]

**Table 13 marinedrugs-18-00142-t013:** Distribution of compounds **474** to **500**.

No.	Compound Type	Species	Origin	Ref
**474**	Bromophenolic	*C. officinale*	Bateau Bay, NSW	[54]
		*P. lucida*	Bateau Bay, NSW	
		*G. secundada*	Batemans Bay, NSW	
		*A. anceps*	Bateau Bay, NSW	
		*J. sagittata*	Bateau Bay, NSW	
		*D. pulchra*	Botany Bay, NSW	
		*S. robusta*	Batemans Bay, NSW	
**475–478**	Bromophenolic	*C. officinale*	Bateau Bay, NSW	[54]
		*P. angustum*	Bateau Bay, NSW	
		*P. lucida*	Batemans Bay, NSW	
		*G. secundada*	Batemans Bay, NSW	
		*A. anceps*	Bateau Bay, NSW	
		*J. sagittata*	Bateau Bay, NSW	
		*D. pulchra*	Botany Bay, NSW	
		*S. robusta*	Batemans Bay, NSW	
**479**	Xanthophyll	*L. filiformis*	Australia	[185]
		*L. botryoides*	Australia	
**480**	Xanthophyll	*L. botryoides*	Australia	[185]
		*M. abscissa*	Leigh, New Zealand	
**481, 482**	Xanthophyll	*A. ciliolatum*	Ensenada, Baja, California	[186]
		*C. clavulatum*	Ensenada, Baja, California	
		*P. capillacea*	Ensenada, Baja, California	
**483**	Xanthophyll	*L. botryoides*	Australia	[185]
**484**	Xanthophyll	*L. botryoides*	Australia	[186]
		*A. ciliolatum*	Ensenada, Baja, California	
		*C. clavulatum*	Ensenada, Baja, California	
		*M. abscissa*	Leigh, New Zealand	
		*P. capillacea*	Ensenada, Baja, California	
**485**	Aromadendrene	*L. filiformis*	South Australia	[127]
**486, 488**	γ−Pyrones	*P. peperocarpos*	South Australia	[178]
**489**	Lipid	*L. filiformis*	Taroona Beach, Hobart, Tasmania	[138]
**490**	γ−Pyrones	*P. australasica*	Pearsons Point, Tasmania	[168]
**491, 492**	γ−Pyrones	*P. australasica*	-	[167]
**493, 494**	Pyrogallols	*G. filicina*	Bay of Hiroshima, Japan	[181]
**495**	Cyclic lipid	*G. filicina*	Bay of Hiroshima, Japan	[182]
**496**	Xanthophyll	*C. clavulatum*	Ensenada, Baja, California	[186]
		*M. abscissa*	Leigh, New Zealand	
		*P. capillacea*	Ensenada, Baja, California	
**497**	Cyclic lipid	*C. clavulatum*	East Coast of Sicily, Italy	[187]
**498**	Xanthophyll	*C. clavulatum*	Ensenada, Baja, California	[186,188]
		*M. abscissa*	Leigh, New Zealand	
**499**	Xanthophyll	*M. abscissa*	Leigh, New Zealand	[188]
**500**	Diterpene	*P. costatum*	Deep Glen Bay, Tasmanina	[142]

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
