# Peer review of "Natural Products of Marine Macroalgae from South Eastern Australia, with Emphasis on the Port Phillip Bay and Heads Regions of Victoria"

_marinedrugs, 2020, doi:10.3390/md18030142_

Round 1

Reviewer 1 Report

The authors review macroalgae and their metabolites present in the South part of Australia. Even if this review can be of interest the studied location seems very restricted and the interest might be very limited. All along the document a lack of rigor can be noticed for example citing genera and species (in italics). Drawings of compounds could be improved and their names written below their structures. Too many structures are found in the si. I believe the authors should devote more efforts to the writing and drawing before this review can be accepted for publication.

The beginning of the introduction is very general. The authors should move quickly to marine natural products and macroalgae. An inventory is not always driven by the pharmacological properties of the metabolites.

L49: authors should consult marine pharmacology website and there are actually 10 marine drugs+1 actually extracted from macroalgae and recently marketed by a Chinese company: oligomannate. There is no need to draw the structures of the compounds marketed in Figure 1!

L60: indeed macroalgae are underrepresented for pharmacological applications so maybe the authors should include current use of algae like food, feed, animal health, fertilisers and cosmetics…

L277 what is a prenylated acetone? Change name

L592. There is no figure caption

Author Response

Reviewer 1: Comments and Suggestions

The authors review macroalgae and their metabolites present in the South part of Australia. Even if this review can be of interest the studied location seems very restricted and the interest might be very limited. All along the document a lack of rigor can be noticed for example siting genera and species (in italics). Drawings of compounds could be improved and their names written below their structures. Too many structures are found in the SI I believe the authors should devote more efforts to the writing and drawing before this review can be accepted for publication.

Response: We thank the reviewer for their comments. The document has been thoroughly checked to ensure that all genera and species have indeed been cited in italics as there were several of these mistakes noted once our initial manuscript was reformatted by the journal. Regarding the structure drawings, it is our view that the document would suffer greatly from overcrowding should all recorded natural products (approx. 500) be displayed within the main body of the manuscript. Due to this, only selected compound structures were included in the manuscript. However, all structures have now been included in the supporting information file so that the reader can locate all structures in the one location. Regarding the naming of compounds, efforts were made, where possible, to name compounds of interest that provided a trivial name, but many of the natural products reported only have IUPAC names which are too long to be conveniently added below the structures. For this reason, only selected natural products, which have either significant biological activity or if they were a chemotaxonomic marker, for instance, were named within the review.

The beginning of the introduction is very general. The authors should move quickly to the marine natural products and macroalgae. An inventory is not always driven by the pharmacological properties of the metabolites.

Response: We agree that inventories are not always driven by the pharmacological properties of metabolites which is why we also were interested in the origin of the secondary metabolites in this review. Having documented the origin of the species for each secondary metabolite, this review also adds value for phycologists who are interested in the origin and location of secondary metabolites occurring in these marine algae species. It is our view that the introduction length is necessary in order to provide suitable context to this review.

L49: authors should consult marine pharmacology website and there are actually 10 marine drugs +1 actually extracted from macroalgae and recently marketed by a Chinese company: oligomannate. There is no need to draw the structures of the compounds marketed in Figure 1!

Response: Thank you for this comment and suggestion. We have checked and now include Enfortumab vedotin and Polatuzumab vedotin to the list of marine approved drugs. We wanted to retain the structures of these compounds in Figure 1 as we feel they are important for the reader of the review to have a quick reference to and that they add value to the final manuscript.

L60: indeed macroalgae are underrepresented for pharmacological applications so maybe the authors should include current use of algae like food, feed, animal health, fertilisers and cosmetics…

Response: The reviewer makes a good point and so this information has been added in the introduction together with references (new lines are 57-59).

L277 what is a prenylated acetone? Change name

Response: We have altered this and now refer to these compounds as monoterpenes or sesquiterpenes.

L592: There is no figure caption

Response: Figure captions have now been added for all chemical structure diagrams in this review.

We thank this review for their valuable suggestions and comments.

Reviewer 2 Report

This review looks at the natural products that have been isolated from the seaweed species found around Port Phillip Bay in Australia. 

It's a comprehensive piece of work and there's some real value in having the collected structures of these natural products drawn across the review and the supplementary information.

I've some comments:

A) Who's the intended readership for this? Is this written by phycologists and aimed at drug researchers, or the other way around? Either way, I think there are some gaps: phycologists are unlikely to understand what distinguishes, say, a monoterpene from a meroditerpenoid and drug researchers are unlikely to know how the various macroalgae are related to each other. This review sheds very little light here: should this be rectified? 

B) Semi-relatedly, this is less of a review and more of a list? It's a pretty comprehensive list, but it's a list. There's some intellectual synthesis both in in the introduction and conclusions, but it's limited in both.

C) To take one example from the introduction: is there any reason why Victorian macroalgae are of anything other than local interest? Can the authors explain what's special about Port Phillip? Is it representative of a particular kind of ecosystem, or is it just where the authors happen to have access to? If some kind of argument can be made here for why PP is worth looking at, it might help to broaden the readership?

D) To take a second example, this time from the conclusions, there'd be some value in a considered discussion of how this work can be used to identify specific algal clades or locations or conditions where we should focus any biodiscovery efforts. Would there be any value to this?

E) What were the methods used for this literature survey?

F) Strictly, this isn't a list of the natural products found in macro algae in Victoria, it's a list of natural products that the species in Victoria have been recorded to be capable of producing (i.e. many reports are from those species but in e.g. Japan). But we know that macroalgal metabolism is environmentally sensitive, so there should be some discussion here: how certain are the authors that the metabolites recorded in, say, Japan are also being produced in Victoria? It might be worth adding a column to all Tables to indicate where the metabolite was actually measured, if different to the location it's been recorded in Victoria?

E) Relatedly, there's very little mention of how much of these products are present in various species? Do some species have lots and others little? This isn't discussed at all, except for (as far as I can see) one instance.

F) How confident are the authors that all macroalgal species found in and around Victoria have been identified?

Minor points:

Title: "with emphasis on": do the authors actually mean "limited to"?

Throughout: Many of the Latin names aren't in italics and should be. Several IC50 values are given without the cells they were ascertained in.

English: A few minor typos. Some species names have been misspelled: linearfolium (p. 14) is linearifolium; variegta (p 15) is variegata; subtilus (p. 16) is subtilis.

Minor points:

p. 2: "advent of SCUBA" Is that an exaggeration? If not, citation?

Fig 2: Scale bar?

Table 1: Perhaps subdivide by phylogeny, rather than alphabetically?

p. 11: MIC is introduced without an explanation.

p. 13: "A recent review" (#72) It's from 1977; it's about as recent as I am...

Table 6: This is all from one paper?

p. 14: "Alga derived from" Probably "Sterols derived from"

p. 21: whereby should be wherein.

Table 11: Make clear it's the Sorrento in Australia, rather than the one in Italy 

Fig 3: Why is this in 3D? Does that not make the values harder to read off the y-axis?

Fig 4: Heatmaps often sort by phylogeny, rather than alphabetically? I've listed this as a minor point, but there's a lot of data in this review and there seems to be an opportunity here to do a thoughtful assessment of the phylogenetic distribution of metabolites (i.e. draw this heatmap by phylogeny): would this not add a lot to the value of this review? 

Fig 4: Also, what's the scale? Also, need Genus names. Also, to what extent is observer bias likely to limit the utility of this heatmap?

Supp Info: Why not put all of the structures in numbered order in the supp info, rather than having the reader have to flip between the paper and the SI? Keep the relevant ones in the paper, but have the full list all in one place in the SI?

Author Response

This review looks at the natural products that have been isolated from the seaweed species found around Port Phillip Bay in Australia. 

It's a comprehensive piece of work and there's some real value in having the collected structures of these natural products drawn across the review and the supplementary information.

I've some comments:

Reviewer 2: Comments and Suggestions

  1. A) Who's the intended readership for this? Is this written by phycologists and aimed at drug researchers, or the other way around? Either way, I think there are some gaps: phycologists are unlikely to understand what distinguishes, say, a monoterpene from a meroditerpenoid and drug researchers are unlikely to know how the various macroalgae are related to each other. This review sheds very little light here: should this be rectified? 

Response: This review is written by natural product chemists interested in the bioactivity and therapeutic potential of the compounds described from marine algae. The intention of the review is to provide an insight into which marine algae species have been studied, to highlight species that have yet to be studied in a phytochemical manner and to provide an overview of the bioactivity reported within these species. The phylogenetic relationship of the marine algae species that have yielded secondary metabolites has now been illustrated in the heat map diagrams (Figures 12-14). The intent of these heat maps is to provide an aid to drug researchers in understanding the close relationship of the marine algae discussed in terms of their phylogeny. When introducing terpene classes, we have now included details on the number of carbons that these structure classes compromise (eg. C-15 for sesquiterpene etc…) which should make it more insightful for a phycologist.

  1. B) Semi-relatedly, this is less of a review and more of a list? It’s a pretty comprehensive list, but it’s a list. There’s some intellectual synthesis both in in the introduction and conclusions, but it’s limited in both.

Response: The review is intended primarily to be a listing of compounds that can be found in the marine algae that can be found in the Port Philip Bay region of Australia. While indeed this represents a listing, we believe this type of review is valuable in that it provides a platform for both natural products chemists and phycologists to quickly view the viability of a large set of marine algae species for research in terms of potential drug development or for further study in terms of secondary metabolite production. We have also extended the discussion in the introduction section of the review to clearly articulate the purpose of the review. We have also added a little to the conclusion to articulate the relevance and importance of the heat maps in showing species diversity in terms of secondary metabolite production.

  1. C) To take one example from the introduction: is there any reason why Victorian macroalgae are of anything other than local interest? Can the authors explain what's special about Port Phillip? Is it representative of a particular kind of ecosystem, or is it just where the authors happen to have access to? If some kind of argument can be made here for why PP is worth looking at, it might help to broaden the readership?

Response: This particular location is ideal for the study of natural products chemistry, in particular macroalgae natural products, due to the intense variety of algal ecosystems found within this Bay. We have added further details of this in the introduction section of the review (Lines 68-76). Specifically, the Port Phillip Bay region is a unique habitat, being shallow enough to be in the photic zone throughout and it is famous for the cleansing activities of the microphyto- and zoo-benthos.   The algae species in the Bay and at the Heads have been sporadically (pre-1960s) to intensively (up to now) sampled and named, and not everything has been studied.  One of the authors (G. Kraft) of this review, together with his students have found and named new genera and species since the 1970s.  Shipping to the Port of Melbourne, coming into the Bay at the Heads, has brought both algal and invertebrate invaders as boats have flushed their ballasts at the entry to the Port, and G. Kraft has described one such alga, a native of the Mediterranean, that became established in Australia via this route.  So the authors declare that there is much to be discovered still in this region.

  1. D) To take a second example, this time from the conclusions, there'd be some value in a considered discussion of how this work can be used to identify specific algal clades or locations or conditions where we should focus any biodiscovery efforts. Would there be any value to this?

Response: This type of information is difficult to present based on the literature available. Much of the literature used for this review references algae that can be found in the Port Phillip Bay region, but have been sampled and studied elsewhere, thus making it difficult to accurately assess viability of specific locations within the Port Phillip Bay region. The intent of this review was to highlight how many of the species for this region would be viable future targets from this region as they have shown to contain bioactive natural products from other localities. Although we acknowledge that secondary metabolism is effected by locality, we still see the value in understanding the types of secondary metabolites that have been found globally. This review can initiate comparative studies where the same species could be compared from different localities. In terms of focusing of biodiscovery efforts, we feel this review provides excellent information as to the species that have already yielded many compounds (see heat maps in Figures 12-14) and thus is useful in directing the reader to a genus that is prolific in secondary metabolites versus one that is not. For example, the genus Laurencia is mentioned frequently in this review due to the large number of secondary metabolites it produces and this highlights the fact that this represents a well-studied species.

  1. E) What were the methods used for this literature survey?

Response: The methodology and purpose of this review has been addressed in the introduction section of this manuscript (Lines 82-89). We have detailed how we obtained our species lists from survey data locations including references to these surveys and Victorian databases. We then also used the SciFinder and MarinLit databases in order to obtain information on each of these marine algae species in terms of secondary metabolite production as reported between the period 1971 to early 2019. We feel this accurately informs the reader about the methodology of the review.

  1. F) Strictly, this isn't a list of the natural products found in macro algae in Victoria, it's a list of natural products that the species in Victoria have been recorded to be capable of producing (i.e. many reports are from those species but in e.g. Japan). But we know that macroalgal metabolism is environmentally sensitive, so there should be some discussion here: how certain are the authors that the metabolites recorded in, say, Japan are also being produced in Victoria? It might be worth adding a column to all Tables to indicate where the metabolite was actually measured, if different to the location it's been recorded in Victoria?

Response: We acknowledge the fact that different localities produce differing secondary metabolites in the introduction of this review. Our aim was to investigate the list of marine algae that were compiled and to then investigate all reports that detailed a phytochemical study. We feel we have addressed this with the “Origin” column in our compound distribution (Tables 2-13). This origin column describes where the organisms discussed were sampled from, giving an indication as to the differences of each locality between similar species.

  1. G) Relatedly, there's very little mention of how much of these products are present in various species? Do some species have lots and others little? This isn't discussed at all, except for (as far as I can see) one instance.

Response: We thank the reviewer for this comment as this is a good point and there is value to be added here. However, this particular information was not the focus of this review. This review was aimed at focusing on the relationship between what secondary metabolites could be found in the marine algae and the associated bioactivities of these compounds.

  1. H) How confident are the authors that all macroalgal species found in and around Victoria have been identified?

Response: This was one of the major sticking points in the process of compiling this review in that there are very few comprehensive taxonomic marine algae species surveys of this area. One of the major tasks for this review was to seek out the one major survey that was performed by H.B.S Womersley in 1959 and try to reconcile this with a more modern survey. We feel we have delivered a list of marine algae species that accurately represents the described and, to date, the known species that inhabit this area through use of the Womersley survey data and the Victorian Government survey data presented in the Victorian Biodiversity Atlas. This list was then cross checked by Dr. Gerald Kraft, who is an expert in marine algae taxonomy of the Port Phillip Bay region, giving us confidence that the marine algae species we have included are indeed from this area. In terms of how sure we are that all marine algae species in this area have been identified, this type of questioning can be applied to any location. However, we have now highlighted in the manuscript (lines 72-75) that the distribution of species in the bay is dynamic due to shipping which can introduce species to the Bay, however, like many locations world-wide, there are still new species being found.

Minor Points:

  1. 2: "advent of SCUBA" Is that an exaggeration? If not, citation?

Response: This was changed to “increased use of SCUBA”

Fig 2: Scale bar?

Response: Scale bar has been added to this figure as requested.

Table 1: Perhaps subdivide by phylogeny, rather than alphabetically?

Response: Thank you for the suggestion but it is our opinion that this table is to be treated almost as an index for the reader and will allow for easier use in the alphabetical order that it is currently in.

  1. 11: MIC is introduced without an explanation.

Response: This has now been introduced with explanation upon first mentioning to indicate “Minimum Inhibitory Concentration (MIC)”.

  1. 13: "A recent review" (#72) It's from 1977; it's about as recent as I am...

Response: Thanks to the reviewer for picking this up as unfortunately this was the incorrect reference and has now been corrected to reflect that it is more recent (2018).

Table 6: This is all from one paper?

Response: Yes, these compounds were compiled in a previous review on this Family of algae.

  1. 14: "Alga derived from" Probably "Sterols derived from"

Response: Sterols was correct, thank you to the reviewer. The change has been made.

  1. 21: whereby should be wherein.

Response: Changed as per reviewer’s suggestion.

Table 11: Make clear it's the Sorrento in Australia, rather than the one in Italy 

Response: This has been made clear by adding “Australia” to the origin section.

Fig 3: Why is this in 3D? Does that not make the values harder to read off the y-axis?

Response: This Figure has been changed to be represented in 2D with solid colours and now also displays values above each bar to provide further clarity.

Fig 4: Heatmaps often sort by phylogeny, rather than alphabetically? I've listed this as a minor point, but there's a lot of data in this review and there seems to be an opportunity here to do a thoughtful assessment of the phylogenetic distribution of metabolites (i.e. draw this heatmap by phylogeny): would this not add a lot to the value of this review? 

Response: We thank the reviewer for their recommendation. Much value can be added to the review by organising the heat map in such a way and so we have changed the heat maps so that the phylogeny of each marine algae species is now apparent in each of these plots.

Fig 4: Also, what's the scale? Also, need Genus names. Also, to what extent is observer bias likely to limit the utility of this heatmap?

Response: The heat maps in this review have now been changed to include both the scale and the genus names of the species. The colour palette for this heat map has been changed to make the zero value white and the highest values dark red to make it easier to interpret. The scale represents the number of compounds for that particular species. The purpose of the heat maps is to clearly illustrate which marine algae species have resulted in the greatest number of secondary metabolites and therefore quickly being able to draw the reader’s attention to understudied marine algae of the area that could be prioritised for future studies.

Supp Info: Why not put all of the structures in numbered order in the supp info, rather than having the reader have to flip between the paper and the SI? Keep the relevant ones in the paper, but have the full list all in one place in the SI?

Response: All of the structures have now been provided in the supporting information. Where the compounds are also mentioned in the review the structures have also been included within the manuscript for quick reference. 

We take the time to thank the reviewer for their comments and suggestions which has improved the manuscript.

Reviewer 3 Report

The manuscript presents a review on the major products which could be obtained from macroalgae from a defined geographical area. Emphasis is given on bioactive products.

A survey of marine derived clinical candidates is presented. Also a comprehensive compilation of compunds and activities is presented in the manuscript and in the Supplementary documentation.

The manuscript presents valuable information and is well organized.

Only some formal revisions are suggested:

Check that Latin names are in italic

Line 35, delete a space before Akin

“Beach” or “beach”

Font size in Table 2

Spelling: “sesquiterpinoids”; “variegta”

Line 234. Prenylated phenolc (68-79) (see Supporting Information Figure S5), but also compound 80 is in this figure

Ref. 42. Delete the name of autor (Maria Antonio)

Author Response

Reviewer 3: Comments and Suggestions

The manuscript presents a review on the major products which could be obtained from macroalgae from a defined geographical area. Emphasis is given on bioactive products.

A survey of marine derived clinical candidates is presented. Also a comprehensive compilation of compounds and activities is presented in the manuscript and in the Supplementary documentation.

The manuscript presents valuable information and is well organized.

Only some formal revisions are suggested:

Check that Latin names are in italic

Response: Although these were correct in the version uploaded, it is apparent that when the manuscript was reformatted by the journal, many of the Latin names that were previously in italics were lost. These have now been thoroughly checked in the latest version of the manuscript and corrected.

Line 35, delete a space before Akin

Response: Change made as per reviewer’s comment.

“Beach” or “beach”

Response: Thank you to the reviewer for picking this up. All locations have now been made consistent using the capitalised “Beach”.

Font size in Table 2

Response: Both font type and size were corrected in Table 2.

Spelling: “sesquiterpinoids”; “variegta”

Response: Spelling was adjusted here to “sesquiterpenoids” and “variegata”.

Line 234. Prenylated phenolic (68-79) (see Supporting Information Figure S5), but also compound 80 is in this figure

Response: This sentence has been updated to include structure 80 as well. Prenylated phenolic (68-80) (see Supporting Information Figure S5). (Now line 251)

Ref. 42. Delete the name of author (Maria Antonio)

Response: This has been changed as requested.

We thank this reviewer for their valuable comments and suggestions.

Comment to editors: We updated the map in Figure 2 in this version of the manuscript.